# Lignin oxidation products as a potential proxy for vegetation and environmental changes in speleothems and cave drip water – a first record from the Herbstlabyrinth, central Germany

Inken Heidke[1], Denis Scholz[2], and Thorsten Hoffmann[1]

[1]Institute of Inorganic Chemistry and Analytical Chemistry, Johannes Gutenberg-University of Mainz, Duesbergweg 10-14, 55128 Mainz, Germany
[2]Institute of Geosciences, Johannes Gutenberg-University of Mainz, J.-J.-Becher-Weg 21, 55128 Mainz, Germany

**Correspondence:** Thorsten Hoffmann (t.hoffmann@uni-mainz.de)

**Abstract.** Here we present the first quantitative speleothem record of lignin oxidation products (LOPs), which has been determined in a Holocene stalagmite from the Herbstlabyrinth Cave in central Germany. In addition, we present LOP results from 16 months of drip water monitoring. Lignin is only produced by vascular plants and therefore has the potential to be an unambiguous vegetation proxy and to complement other vegetation and climate proxies in speleothems. We compare our results with
stable isotope and trace element data from the same sample. In the stalagmite, LOP concentrations show a similar behaviour as P, Ba and U concentrations, which have previously been interpreted as vegetation proxies. The LOP ratios S/V and C/V, which are usually used to differentiate between angiosperm and gymnosperm and woody and non-woody vegetation, show complex patterns suggesting additional influencing factors, such as transport and microbiological effects. The drip water from a fast drip site shows a seasonal pattern of LOPs with low LOP concentrations in winter and higher LOP concentrations in summer.
These results indicate the potential of LOPs as a new proxy for vegetational and environmental changes in speleothems, but also demonstrate the complexity and the current limitations of our understanding of the transport of lignin from the soil into the cave and the speleothems.

## 1 Introduction

Speleothems are valuable climate archives because they can grow continuously for thousands of years and can be dated accurately 640,000 years back in time using the $^{230}$Th-U method (Cheng et al., 2016; Richards and Dorale, 2003; Scholz and Hoffmann, 2008). Furthermore, the cave provides a preservative environment that protects the recorded chemical proxy signals against outer influences such as light, abrupt changes in temperature and, under ideal conditions, also mechanical disturbance. Until recently, mostly stable isotope ratios and trace elements have been used as climate proxies in speleothems (McDermott,
2004; Fairchild and Treble, 2009). The methodology is well-established, but the records sometimes cannot be interpreted with-

out doubt, unless other proxies are available for comparison (Lachniet, 2009; Fairchild and Treble, 2009; Mischel et al., 2017; Scholz et al., 2012). Therefore, it is important to expand the proxy toolbox of paleoclimate and especially paleo-vegetation reconstruction. The calcite crystal is able to incorporate and thus preserve not only trace elements and stable isotope ratios, but also organic molecules.

Organic matter in speleothems and cave drip water has most often been analyzed as total organic carbon (TOC) or by fluorescence spectroscopy (Quiers et al., 2015). Size exclusion chromatography coupled to organic carbon detection (LC-OCD) was applied to divide organic matter in cave drip water in different fractions, such as biopolymers and humic substances (Rutlidge et al., 2015). Only very few molecular organic analytes have been analyzed in speleothems so far. Glycerol dialkyl glycerol tetraethers (GDGTs), which are produced by microorganisms in situ in the cave or the overlaying vadose zone, have

been implemented as organic temperature proxies (Blyth et al., 2013, 2014). Lipid biomarkers, such as fatty acids (Bosle et al., 2014) and especially long chain *n*-alkanes from plant leave waxes, have been used as vegetation proxies, and there have been approaches to use the chain length distribution of *n*-alkanes to distinguish between the input of grasses and woody plants (Xie, 2003; Blyth et al., 2007, 2011). However, there are uncertainties about the validity of chain length distributions to distinguish between different plant groups (Bush and McInerney, 2013; Blyth et al., 2016). In addition, lipid biomarkers are especially

prone to laboratory contamination, which poses a general problem for biomarker analysis (Wynn and Brocks, 2014). Therefore, a more specific plant biomarker is needed that is less prone to contamination.

Lignin is one of the main constituents of wood and woody plants. In the soil, lignin contributes to soil organic matter and can be found in the alkaline extracts traditionally referred to as humic substances, mainly in the humic acids fraction (Kögel-Knabner, 2002; Lehmann and Kleber, 2015). The microbial degradation of lignin is comparably slow, as only white-rot fungi

are able to completely mineralize it to $CO_2$ in co-metabolism with other energy sources, whereas some other microorganisms are only able to induce structural changes to lignin (Kögel-Knabner, 2002). Although the exact fate of lignin in soils is still a matter of debate (Thevenot et al., 2010), its relative recalcitrance has led to lignin oxidation products being widely used as paleo-vegetation proxy in archives, such as peat, lake sediments and marine sediment cores (see, for instance, the review by Jex et al., 2014) and as a proxy for terrestrial input of plant biomass in natural waters, like rivers and oceans (e.g., Zhang

et al., 2013; Standley and Kaplan, 1998; Hernes and Benner, 2002). Blyth and Watson (2009) first detected lignin phenols in speleothems, and Blyth et al. (2008) and Blyth et al. (2016) have highlighted lignin oxidation products as promising vegetation proxies in speleothems. Recently, Heidke et al. (2018) developed a method to quantitatively analyze lignin as a paleo-vegetation proxy in speleothems and cave drip water. The advantage of lignin as a vegetation proxy is that it is only produced by vascular plants and not by microorganisms. Thus, it is very specific, and the risk of laboratory contamination is also much lower than for

ubiquitous substances, such as lipids. In addition, lignin analysis does not only give information about the abundance, but also about the type of vegetation. The biopolymer lignin consists of three different monomers, coniferyl alcohol, sinapyl alcohol and p-coumaryl alcohol. The proportion of these three monomers varies with the type of vegetation. Lignin from gymnosperm wood mainly consists of coniferyl alcohol, whereas lignin from angiosperm wood contains coniferyl and sinapyl alcohol. Non-woody vegetation, like grasses, leaves and needles, is characterized by a higher proportion of p-coumaryl alcohol and

ester-bound p-coumaric acid and ferulic acid (Boerjan et al., 2003). To analyze the lignin composition, the lignin polymer has

to be oxidatively degraded into monomeric lignin oxidation products (LOPs). The guaiacyl phenylpropanoid (from coniferyl alcohol) is oxidized to vanillic acid, vanillin and acetovanillone (V-group LOPs), the syringyl phenylpropanoid (from sinapyl alcohol) to syringic acid, syringaldehyde and acetosyringone (S-group LOPs), and the p-hydroxyphenyl phenylpropanoid (from p-coumaryl alcohol) to p-hydroxybenzoic acid, p-hydroxybenzaldehyde and p-hydroxyacetophenone (H-group LOPs). Since the H-group LOPs can also originate from other sources than lignin, such as soil microorganisms or the degradation of protein-rich material, they are usually not used as vegetation proxies. The C-group LOPs consist of ferulic acid and p-coumaric acid. Usually, the sum parameter $\Sigma 8$, which is the sum of all eight individual LOPs from the C-, S- and V-group, is used to present the total lignin concentration. In addition, the ratios of the different LOP groups, C/V and S/V, are used to present the type of lignin, where a higher C/V ratio indicates a higher contribution of non-woody vs. woody plant material and a higher S/V ratio indicates a higher contribution of angiosperm vs. gymnosperm plant material (Hedges and Mann, 1979).

It is still a subject of research to what extent and how these ratios are affected by factors such as transport, microbial transformation, soil characteristics and land use as well as the interaction with mineral surfaces (Hernes et al., 2007; Thevenot et al., 2010; Hernes et al., 2013; Jex et al., 2014). A clarification of all these aspects is beyond the scope of this study. The aim of this study is to present first results on the application of the lignin analysis method developed by Heidke et al. (2018) to speleothem and cave drip water samples to generally evaluate the potential of LOPs as a proxy for vegetational and environmental changes in speleothem archives. We present the first quantitative record of lignin oxidation products from a Holocene speleothem from the Herbstlabyrinth in central Germany. Our objectives were (i) to investigate, if the sensitivity of our method is high enough to detect and quantify LOPs with a sufficient temporal resolution to reveal centennial to millennial climate changes, and (ii), if and how the LOP signal in speleothems varies with climatic and vegetational changes and how it compares with other, established proxy signals, such as trace elements and stable isotopes. Therefore, we compared our LOP results with trace element and stable isotope records from the same sample (Mischel et al., 2017). In addition, we aimed to improve our knowledge on how lignin is transported into the cave, which is a key question in understanding LOPs as an environmental proxy in speleothem archives. Therefore, we investigated seasonal variations of LOP concentrations in drip water from the same cave, sampled monthly over a period of 16 months in the framework of a cave monitoring program.

We chose the Herbstlabyrinth Cave for our first quantitative LOP analysis because of an ongoing cave monitoring program, and the specific Holocene stalagmite was chosen because it was already well characterized by stable isotope and trace element analysis, the general vegetation and climate changes in the Holocene in central Germany are relatively well known (e. g. Litt et al., 2009), and last but not least, the fast growth rate of the stalagmite offered sufficient sample material to test and apply our method.

## 2 Materials and methods

### 2.1 The cave monitoring program in the Herbstlabyrinth

From 2010 to 2015, a monthly cave monitoring program, including drip water sampling and the measurement of cave air temperature and pCO$_2$ as well as meteorological data, was conducted at the Herbstlabyrinth. A detailed description of the
5 monitoring program, the cave and its environment can be found in Mischel et al. (2015) and Mischel et al. (2017). In brief, the Herbstlabyrinth is situated in the Rhenish Slate Mountains in central Germany. The cave system is about 11 km long and well decorated with different kinds of speleothems. The vegetation above the cave consists of deciduous forest and grassland, and the soil above the cave is a 60 cm-thick Cambisol (Terra fusca). The mean annual temperature at the cave site is 9.0 °C, and the mean annual precipitation is around 800 $mm \cdot a^{-1}$, with evenly distributed rainfall throughout the year. Due to higher
evapotranspiration during summer, the recharge of the aquifer and thus the drip water mainly consists of winter precipitation, but heavy rainfall events in summer can also have a substantial contribution (Mischel et al., 2015). The drip water samples used for lignin analysis were sampled monthly from May 2014 until August 2015. They were collected in precleaned glass vessels from one fast drip site (*D1*, average drip rate approx. 0.3–0.5 drops per second), one slow drip site (*D2*, average drip rate approx. 60 mL per month) and one cave pool (*PW*). To prevent the growth of microorganisms, 5% (w/w) of acetonitrile
was added to the samples, which were then stored in the dark at 4 °C for several months.

### 2.2 The stalagmite sample

Stalagmite NG01 has a light whitish to yellowish colour, is 50 cm long, has a diameter of 15 cm, and grew during the Holocene (Mischel et al., 2017). It was dated via the [230]Th–U method using multicollector-ICP-MS (Mischel et al., 2017). To calculate the age-depth model, the algorithm StalAge was used (Scholz and Hoffmann, 2011). The stalagmite samples used for lignin
analysis were cut from a 1 cm thick slab following visible growth lines and had a width along the growth axis of 0.5–2.0 cm, a length of 1.2–4.8 cm and a weight of 2.4–6.2 g. To determine the age and the corresponding error of these samples, the calculated ages for the mid depth and the depth of the upper and lower edges of the sample were used.

### 2.3 Analytical methods

The analytical method for the analysis of LOPs is described in detail in Heidke et al. (2018). In brief, the stalagmite samples for
LOP analysis were cleaned with organic solvents, edged on the outside with diluted HCl to prevent the influence of potential laboratory contamination (Wynn and Brocks, 2014), and finally dissolved in ultra pure 30% HCl. These sample solutions as well as the drip water samples were extracted via solid phase extraction (SPE), and the extracted lignin was degraded via microwave-assisted CuO oxidation. The resulting lignin oxidation products (LOPs) were again extracted via SPE and then analyzed via ultrahigh-performance liquid chromatography (UHPLC) coupled to heated electrospray ionization (HESI)
high resolution mass spectrometry (HRMS) using a Dionex Ultimate 3000 UHPLC system and a Q-Exactive Orbitrap mass

spectrometer (Thermo Fisher Scientific). A detailed discussion of possible blank contaminations and measures to prevent these is given in Heidke et al. (2018).

The methods used for dating of the stalagmite samples, the analysis of trace elements and stable isotopes as well as the calculation of the drip rate for the cave drip water samples and the growth rate of the stalagmite are described in detail in Mischel et al. (2017) and references therein. To compare the LOP results with stalagmite data of stable isotopes and trace elements, which have a much higher resolution of 2 mm per sample, a mean value of the higher-resolution data according to the sample size of the LOP samples was calculated, and the standard deviation of these mean values was used as error-bars. In the discussion, only these lower-resolution data will be shown and discussed. The original data can be found in Mischel et al. (2017). For the drip water stable isotope and trace element data, the uncertainty is very small and therefore not shown here, but it is provided in Mischel et al. (2017). For the drip water LOP data, the uncertainty was calculated as described in Heidke et al. (2018), and for the stalagmite LOP data, the calculation of the uncertainty is described in section 3.1.

## 3 Results

### 3.1 LOPs in stalagmite samples

One complication of studies with a large number of samples that have to be analyzed over an extended period of time are so-called batch effects, which occur because measurements are affected by laboratory conditions, such as reagent lots, instrumental drift, stability of the reference standards or a varying efficiency of the sample preparation steps (e.g., $CuO$ oxidation step). Since such effects are almost unavoidable, adjustment strategies are generally required (Wehrens et al., 2016; Kirwan et al., 2013; Surowiec et al., 2017), which is especially true when temporal records of the target analytes are the aim of the study. To do so, the stalagmite samples for LOP analysis were analyzed in six batches with nine samples and one blank sample per batch. To recognize systematic errors leading to batch effects, the samples within each batch were not located side by side in the stalagmite, but evenly distributed over the whole length of the stalagmite. In fact, in the results of the LOP concentrations and the C/V and S/V ratios, we observed a regular pattern, which corresponded to the different batches. To determine the original signal and overcome the differences in instrumental response between the batches, the results of the individual measurements were revised based on the following correction procedure. In the oldest part of the stalagmite (11.2–8.6 ka BP), the pattern was most visible and least superimposed by original signals. Therefore, we used only this part to calculate a correction factor using equation (1),

$$x_{\text{smoothed}} = x - \bar{X}_{\text{batch k}} + \bar{X}, \tag{1}$$

with $x_{\text{smoothed}}$ representing the corrected (smoothed) value of $\Sigma 8$, C/V or S/V, $x$ representing the original value of $\Sigma 8$, C/V or S/V, $\bar{X}_{\text{batch }k}$, $k = 1, 2, ..., 6$, the mean value of all $x_i$ in batch $k$, and $\bar{X}$ the mean value of all $x_i$ of all batches in the oldest part of the stalagmite. The error bars $\Delta x_{\text{smoothed}}$ for the smoothed values were calculated by equation (2),

$$\Delta x_{\text{smoothed}} = \sqrt{(\Delta x)^2 + (\Delta \bar{X}_{\text{batch k}})^2 + (\Delta \bar{X})^2}, \tag{2}$$

with $\Delta \bar{X}_{\text{batch k}}$ and $\Delta \bar{X}$ representing the standard deviations of the respective mean values and $\Delta x$ representing the uncertainty of a single sample analysis as described in Heidke et al. (2018). The smoothed and original data for $\Sigma 8$ are shown in Fig. 1, and for C/V and S/V in Fig. 2.

Figure 1 shows the concentrations of the C-, S- and V-group LOPs as well as the $\Sigma 8$ concentrations plotted against the age of the stalagmite. Figure 2 shows the C/V and S/V ratios. The ratios of vanillic acid to vanillin (Vac/Val) and syringic acid to

10 syringaldehyde (Sac/Sal) can be seen in Fig. S2 in the SI.

The mean $\Sigma 8$ concentration in the whole stalagmite was $51 \pm 15 \text{ ng} \cdot \text{g}^{-1}$. The highest concentrations of up to $92 \text{ ng} \cdot \text{g}^{-1}$ occurred shortly after the hiatus at the beginning of the middle part at 7.6 ka BP, followed by the lowest concentrations with $19 \text{ ng} \cdot \text{g}^{-1}$ at 7.0 ka BP. The mean C/V ratio was $0.32 \pm 0.15$, with the lowest ratio of 0.12 at 10.9 ka BP and the highest ratios of up to 0.68 at 0.3, 6.5 and 7.2 ka BP. The mean S/V ratio was $2.15 \pm 0.71$, with the lowest ratio of 1.08 at 11.1 ka BP and the

15 highest ratios of up to 4.18 at 5.1 and 7.1 ka BP.

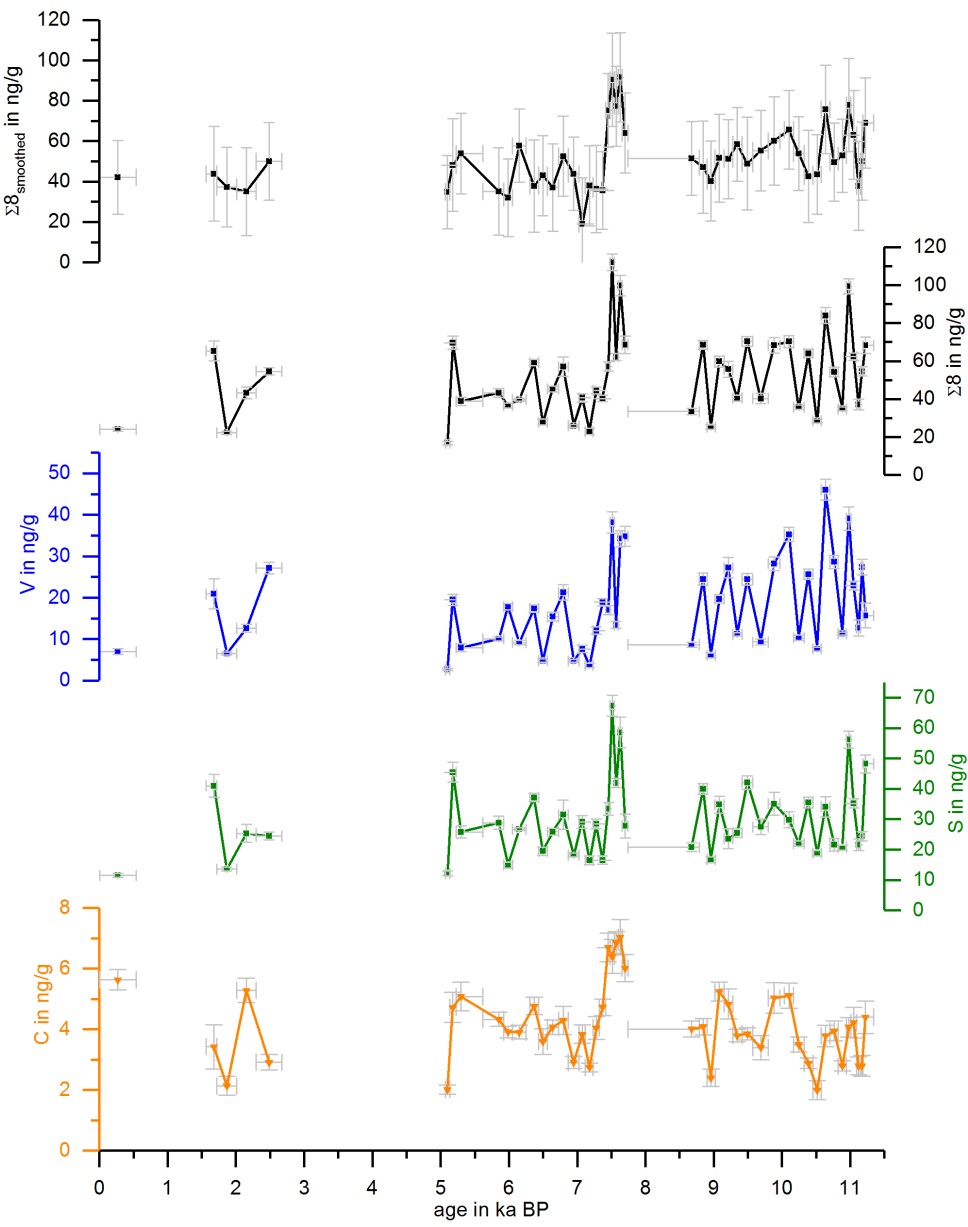

**Figure 1.** Concentrations of the C-, S- and V-group LOPs and Σ8 concentrations plotted against the age of the stalagmite. The uppermost plot shows the smoothed results of Σ8 concentrations.

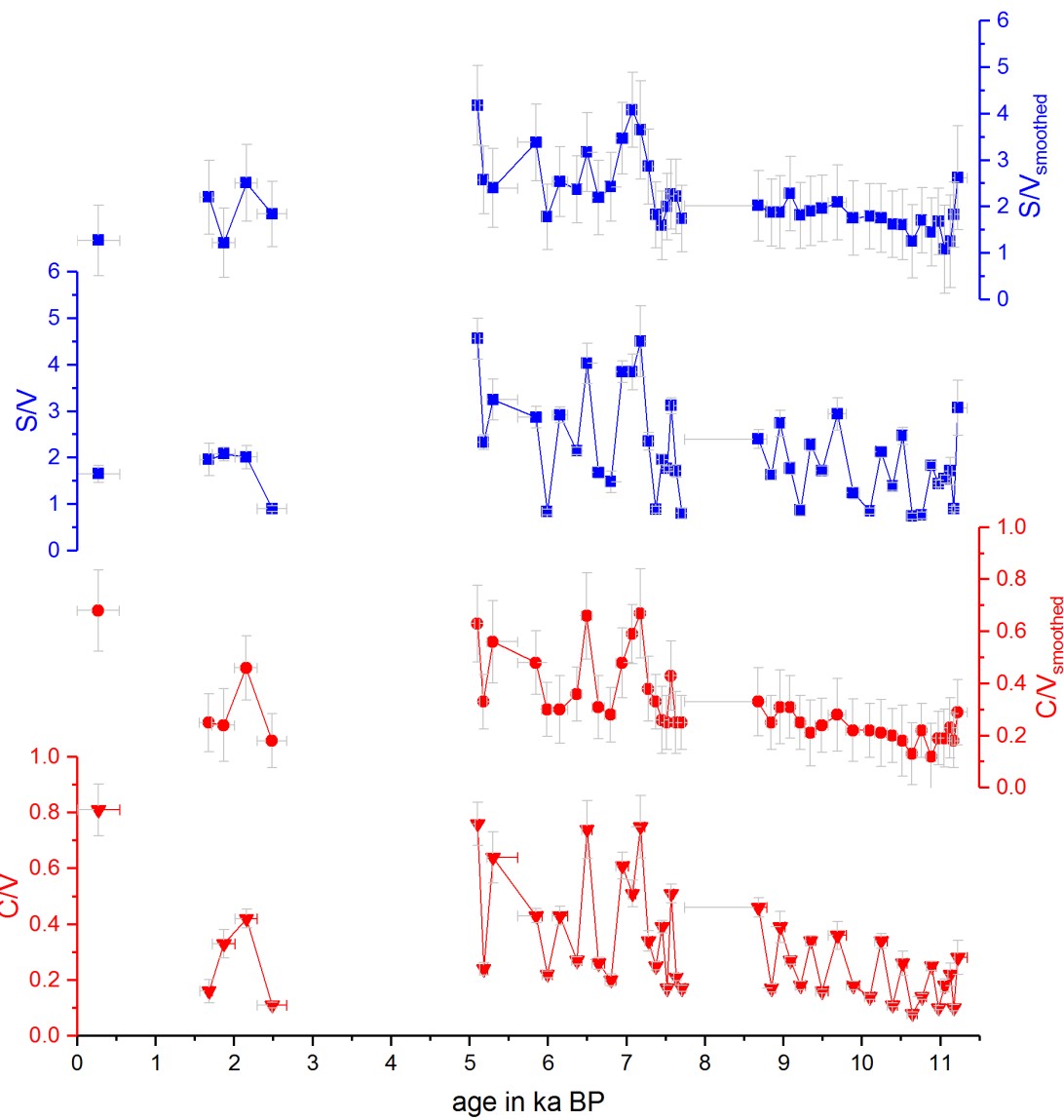

**Figure 2.** Ratios of C/V (red) and S/V (blue), original data and smoothed data, plotted against the age of the stalagmite.

## 3.2 LOPs in drip water samples

In Fig. 3, the LOP results of the fast drip site, *D1*, the slow drip site, *D2*, and the cave pool water, *PW*, are shown. The height of the columns represents the sum of the LOP concentrations, $\Sigma 8$, and the different colors show the contribution of the C-, S- and V-group LOPs. In general, the V-group LOPs contribute the largest part to the total LOP concentration, followed by the S-group and the C-group LOPs. *D1* shows a strong seasonal pattern with high LOP concentrations between 500 and 1800 $\mathrm{ng \cdot L^{-1}}$ from May to August 2014 and from April to August 2015, and low LOP concentrations between 30 and 400 $\mathrm{ng \cdot L^{-1}}$ from September 2014 to February 2015. For July 2014 and March 2015, no data are available. For the slow drip site, *D2*, the concentrations do not show a seasonal pattern, but are highly variable between 80 and 3000 $\mathrm{ng \cdot L^{-1}}$. In May, July and August 2014 and June, July and August 2015, there are no data available, mainly because the drip rate was too low to collect a sufficient sample volume. In the pool water, *PW*, the LOP concentrations show a similar seasonal pattern as in the fast drip site, but less pronounced with concentrations between 140 and 650 $\mathrm{ng \cdot L^{-1}}$ from September 2014 to February 2015 and between 440 and 1100 $\mathrm{ng \cdot L^{-1}}$ in the other months, with one exception in April 2015 with more than 2500 $\mathrm{ng \cdot L^{-1}}$.

In Fig. 4, the C/V and S/V ratios are shown for *D1*, *D2* and *PW*. Because concentrations of individual analytes were below the limit of detection, the ratios could not be calculated for every sample. For the different drip sites, the ratios are in a similar range. The ranges of the ratios in *D1* are 0.02–0.33 for C/V and 0.14–1.21 for S/V. In D2, the ranges are 0.00–0.20 for C/V and 0.08–0.68 for S/V. In *PW*, the ranges are 0.00–0.43 for C/V and 0.18–0.77 for S/V. The ratios of vanillic acid to vanillin (Vac/Val) and syringic acid to syringaldehyde (Sac/Sal) are shown in Fig. S1 in the supplementary information (SI).

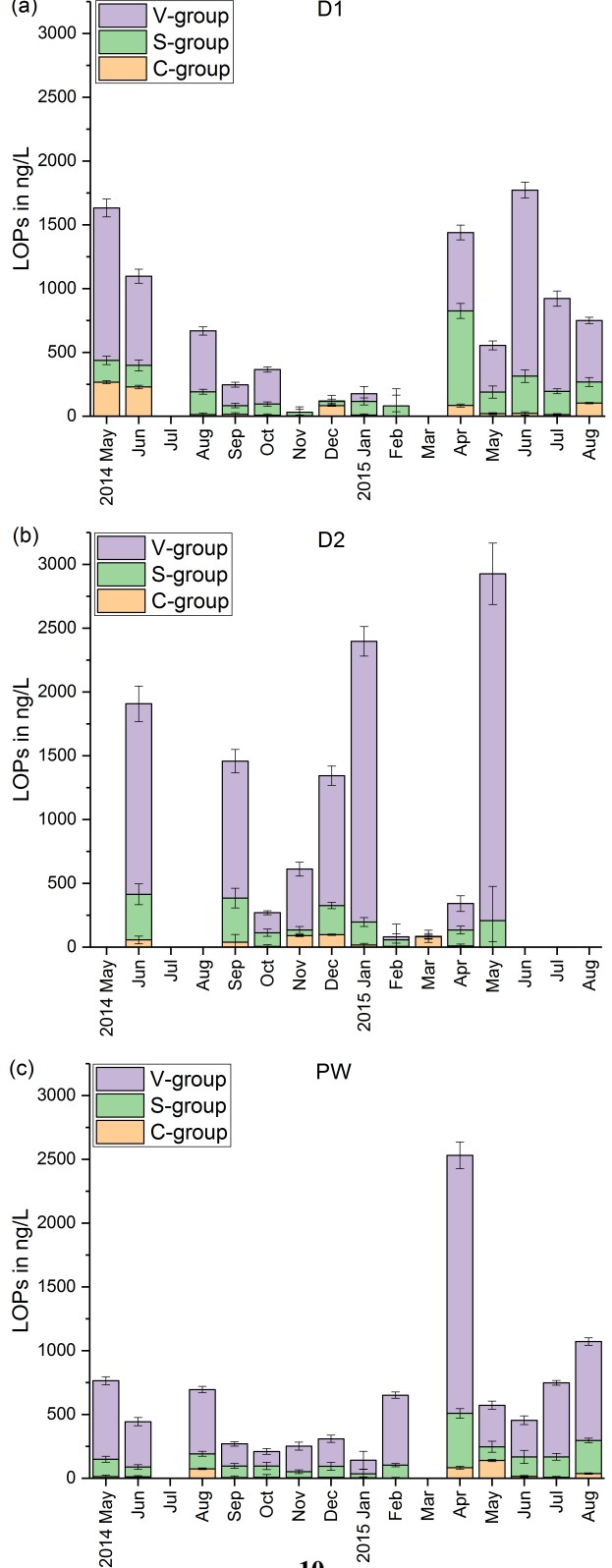

**Figure 3.** LOP results in drip water. (a) fast drip site *D1*, (b) slow drip site *D2*, (c) cave pool water. The height of the columns represents the sum of the LOP concentrations, Σ8, and the different colors show the contribution of the C-, S- and V-group LOPs.

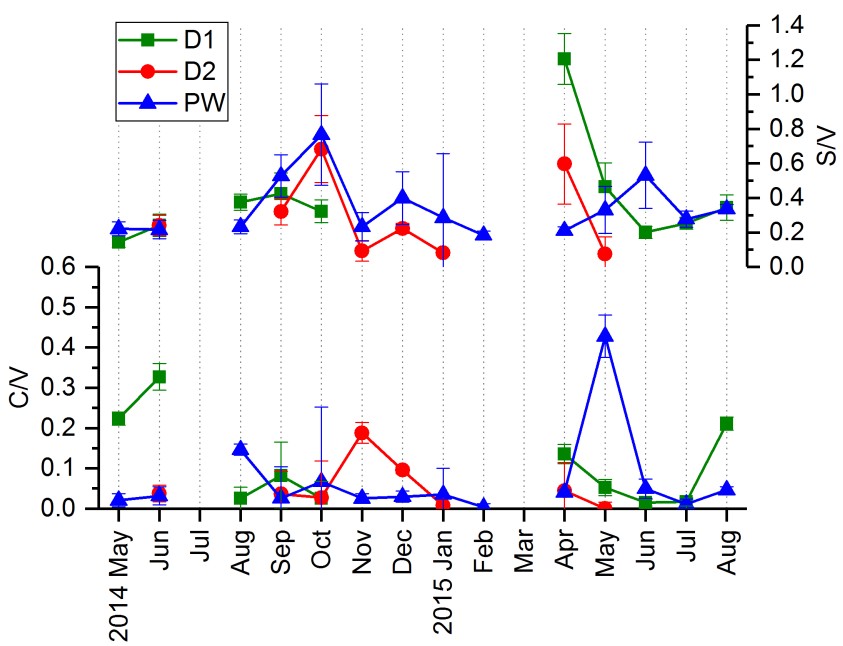

**Figure 4.** C/V and S/V ratios in the drip water.

## 4 Discussion

### 4.1 Stalagmite samples

First of all, we can state that LOPs were detectable in all stalagmite samples and above the quantification limit. Moreover, the signals show a variation over time on the centennial to millennial scale. In Fig. 5, the lignin parameters $\Sigma 8$, C/V and S/V of the stalagmite samples are compared with several trace elements, stable isotopes and the growth rate of the stalagmite. To get an overview of the correlations between the LOPs and the trace element and stable isotope data, we performed a principal component analysis (PCA).[1] As the three growth phases of the stalagmite show a different behaviour in several proxies and the growth rate (Mischel et al., 2017), the PCA and the correlation coefficients were calculated separately for each growth phase. As the youngest part consists of only five LOP samples, we focus the discussion on the middle and older part. (The contribution coefficients of the PCA are shown in Table S7 in the SI, the eigenvalues and percentages of variance in Table S8 in the SI, and all correlation coefficients of both Pearson's linear correlation and Spearman's rank correlation are shown in Tables S1 to S6 in the SI.)

The PCA for the middle part is shown in Fig. 6. In the middle part, principal component 1 (PC1) explains 45.0% of the overall variance and consists mainly of P, Ba, U and $\Sigma 8$ with positive contributions and C/V, S/V and $\delta^{13}$C with negative contributions. Mischel et al. (2017) interpreted P, Ba and U in the Herbstlabyrinth as vegetation proxies, with higher concentrations of these elements indicating a more productive vegetation, coinciding with wetter climate conditions. Since lignin has its source unambiguously in the vegetation, the correlation of $\Sigma 8$ with P, Ba and U supports this interpretation. $\delta^{13}$C values were interpreted as being at least partially influenced by soil $pCO_2$ and soil thickness and thus indirectly affected by vegetation changes (Mischel et al., 2017). PC2 explains 23.3% of the variation and consists mainly of $\delta^{18}$O, Mg and the growth rate on the positive side (with smaller contributions of $\Sigma 8$) and, with smaller coefficients, U, C/V and S/V on the negative side. The fact that $\Sigma 8$ and C/V and S/V appear in both PC1 and PC2 indicates that the $\Sigma 8$ concentration and especially the C/V and S/V ratios are not only influenced by the abundance and type of vegetation, but also by hydrological and soil microbiological effects, such as the transport of organic matter through the soil and the karst system.

The PCA for the older part is shown in Fig. 7. In the older part of the stalagmite, PC1 explains 51.1% and consists mainly of C/V, S/V, U and Ba on the positive side and Sr and Mg on the negative side. It can be inferred from the scores of the individual samples that the influence of Sr and Mg is mainly dominant in the older part of the stalagmite. This long-term decrease of Sr and Mg (see Fig. 5) was interpreted by Mischel et al. (2017) as the result of a thin loess cover, deposited during the last Glacial, being progressively leached at the beginning of the Holocene. PC2 explains 20.4% and consists mainly of positive contributions of the growth rate and negative contributions of $\delta^{13}$C and $\delta^{18}$O. $\Sigma 8$ only appears in the third principal component (explaining 11.4%), with a high positive contribution, together with positive contributions of P (see Table S7 in the SI). This suggests that in the older part of the stalagmite, the influence of vegetation changes on the speleothem signals only plays a subordinate role.

---

[1]The PCA is based on Pearson's linear correlation. In contrast to Mischel et al. (2017), we did not detrend the records before calculating the correlations.

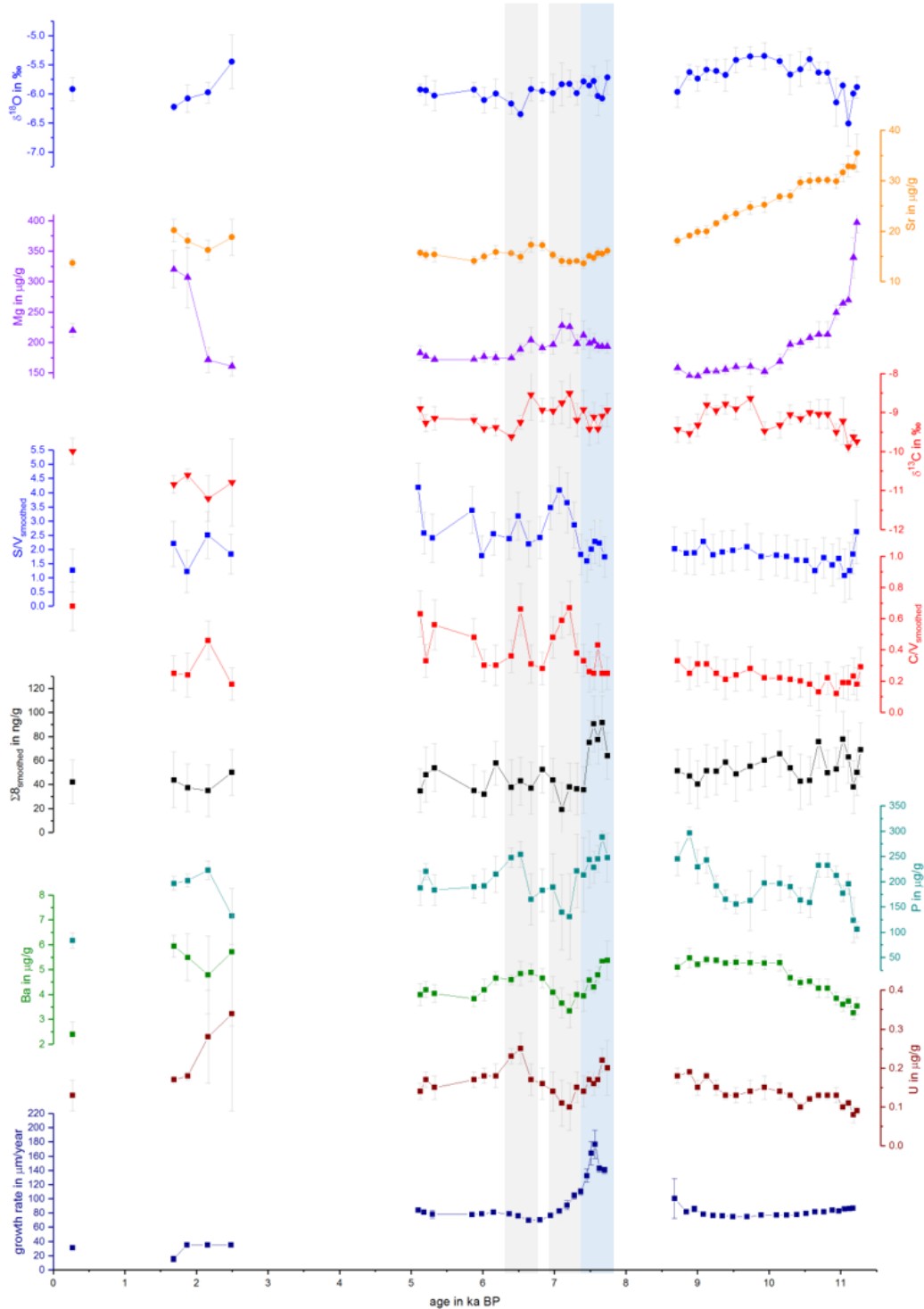

**Figure 5.** Comparison of stable isotopes, trace elements, Σ8 and C/V and S/V ratios, plotted against the age of the stalagmite. The grey bars highlight selected peaks in the C/V and S/V records and the blue bar a peak in the growth rate, which are discussed in the text.

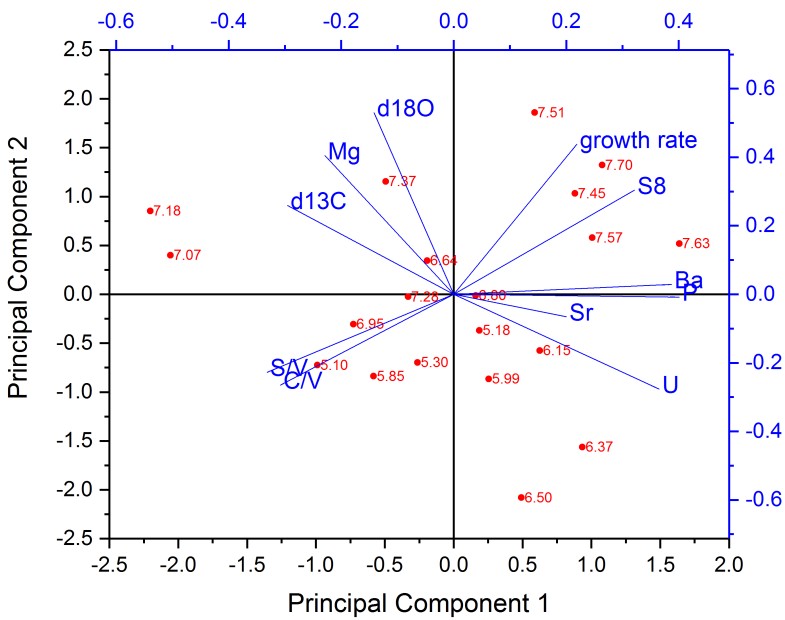

**Figure 6.** Principal component analysis for the middle part of the stalagmite.

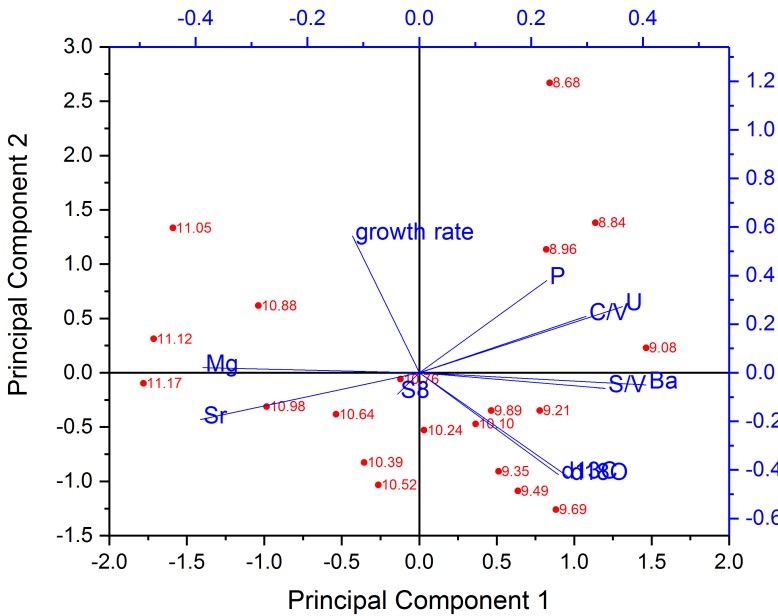

**Figure 7.** Principal component analysis for the older part of the stalagmite.

Taking a closer look at the records in Fig. 5, there is a peak in the growth rate at the beginning of the middle part (around 8.0–7.5 ka BP, blue bar in Fig. 5) that is accompanied by a peak in $\Sigma 8$ and also in P, Ba, and U concentrations. A higher growth rate can be caused, inter alia, by a higher drip rate or by a higher supersaturation of the drip water with respect to $CaCO_3$ (Dreybrodt and Scholz, 2011; Fairchild and Baker, 2012). The latter, in turn, can be caused by a higher soil $pCO_2$ due to a

more productive vegetation and higher microbial activity in the soil, which would be in accordance with the observed higher concentrations of all vegetation proxies. On the other hand, this peak in the growth rate and the vegetation proxies occurs shortly after a hiatus. Therefore, it is also possible that a change in the flow path of the drip water feeding the stalagmite was responsible for the faster growth of the speleothem and a higher input of organic matter. To further investigate this kind of questions, it might be useful in the future to combine LOP analysis with total organic carbon measurements and fluorescence

spectroscopy to gain more information on the amount and composition of organic matter in the speleothem.

The C/V and S/V records have a distinct positive peak at around 7.2 ka BP, and a second, less distinct, positive peak at 6.5 ka BP (grey bars in Fig. 5). At approximately the same time, the concentrations of $\Sigma 8$ and P are lower or even show negative peaks, indicating less input of organic material. Furthermore, the records of $\delta^{13}C$ and Mg show positive peaks at approximately 7.2 and 6.6 ka BP. Higher concentrations of Mg coinciding with more positive $\delta^{13}C$ values in speleothems

are often associated with prior calcite precipitation and usually indicate drier conditions (Mischel et al., 2017; Fairchild and Baker, 2012). Consequently, these peaks could be interpreted as drier periods with less input of organic material. Following the widespread interpretation of C/V and S/V as vegetation source indicators, a possible interpretation of the C/V and S/V records is therefore that during drier times with generally lower lignin input, there was less woody and more non-woody plant material available as a lignin source and the vegetation possibly consisted more of grasses and shrubs. Vice versa, at times

with generally higher lignin input, the lignin consisted of more woody material and the vegetation probably consisted more of forest. However, biotic factors in the soil, such as different rates of microbial degradation for the different types of lignin, as well as transport factors in the soil or the aquifer could also have an influence on the C/V and S/V ratios. Degradation studies of leaf litter showed that the microbial degradation of lignin high in C- and S-group monomeric units is faster than for lignin high in V-group units due to structural differences between the monomers (Bahri et al., 2006; Opsahl and Benner, 1995; Jex

et al., 2014). In addition, adsorption of lignin to mineral particles can lead to an increase in the C/V and S/V ratios, indicating a preferential adsorption of lignin high in V-group monomeric units (Hernes et al., 2007, 2013). This may happen both in the soil and the aquifer. We assume that more degraded lignin is present in smaller fragments and a higher oxidation state and should therefore be better soluble in water. In addition, we hypothesize that this could lead to a more efficient transport through the soil and the aquifer to the cave. This could explain the increase of C/V and S/V ratios during drier times. In addition, the transport

of lignin is also likely to involve processes such as coiling with humic substances in the soil. Currently, we cannot quantify the contribution of the individual mentioned processes to the final lignin signals in the drip water and speleothem calcite. To further investigate all these influencing factors, comparative LOP studies of soil, drip water and speleothem samples need to be conducted.

The pollen-based climate reconstruction by Litt et al. (2009) from the Meerfelder Maar and the Holzmaar in the Eifel region,

about 120 km from the Herbstlabyrinth, shows a relatively stable vegetation development without any abrupt changes, at least

not at the times observed in the C/V and S/V record (6.6 and 7.2 ka BP). This could be interpreted in different ways. One interpretation is that the assumed vegetation changes were local to the region of the the Herbstlabyrinth and did not affect the Eifel region. Another interpretation could be that our proxies are more sensitive to vegetation changes than the pollen record. And third, it is possible that the changes in the vegetation proxies and C/V and S/V were rather caused by changes in the flow paths of the drip water or by site specific soil microbiological processes than by changes in the overlaying vegetation. However, Litt et al. (2009) state that their reconstruction method rather underestimates than overestimates climate changes. To further study the potential of LOPs as a vegetation proxy in the future, it would be beneficial to use speleothem samples from an epoch with more pronounced vegetation changes.

## 4.2 Drip water samples

To gain a better understanding of the transport mechanisms and preservation of LOPs in the cave system, we complemented our first stalagmite LOP record with the analysis of LOPs in monthly sampled drip water (Fig. 3). The study of LOP concentrations over the course of a hydrological year can yield useful information on transport processes and fluxes on a seasonal timescale, provided that the influence of a prolonged dwelling time in the epikarst zone is taken into account. By comparison of modeled and measured $\delta^{18}O$ values, Mischel et al. (2015) have calculated that the drip water of the Herbstlabyrinth is fed by a large reservoir in the karst aquifer, where the water has a residence time of approx. 10 months and consists of rain water mixed over a period of ca. 12 months. Therefore, seasonal variations are strongly attenuated, but should not be shifted much.

In the fast drip site, *D1*, a strong seasonality with higher $\Sigma 8$ concentrations in summer and lower concentrations in winter was observed (Fig. 3). One possible reason could be the growth season of the vegetation from spring to autumn and the activity of soil microorganisms during the warmer time of the year, which would lead to a higher abundance of – partially degraded and therefore transportable – lignin in the soil. However, the degradation of lignin in the soil takes months to years (Thevenot et al., 2010; Bahri et al., 2006). In combination with the long residence and mixing times of the drip water, the original seasonality of the input of plant material on the surface will probably be smoothed out. It seems more likely that the observed seasonal signal is caused by hydrological effects. A possible reason could be dilution of the – otherwise relatively constant – lignin concentration from the reservoir of organic matter in winter, when the recharge of the aquifer and consequently the drip rate are higher (blue bars in Fig. 9 and Fig. 11). In summer, in contrast, the slower drip rate and lower recharge of the aquifer might cause a higher concentration of the organic matter and therefore the lignin content in the drip water (Fig. 8). A combination of both factors – the activity of the vegetation and the soil microorganisms in summer and the dilution effect due to higher recharge of the aquifer in winter – is also possible to explain the observed seasonal concentration pattern of $\Sigma 8$. For future experiments, it might be more useful to apply long-term sampling devices, such as in-situ preconcentration with solid-phase extraction cartridges, instead of whole-water sampling once per month. This would allow quantification of the input of lignin per month instead of the concentration per liter drip water.

Interestingly, Bosle et al. (2014), who analyzed low-molecular weight saturated fatty acids as biomarkers for microbial activity in drip water samples from the same cave monitoring program, found a similar seasonal trend with high concentrations during the summer months and low concentrations in winter for the longest fatty acid, arachidic acid ($C_{20}$), but the opposite

trend for the shorter chained fatty acids $C_{12}$ to $C_{18}$. Their interpretation was that $C_{20}$ could be derived from higher plants above the cave, whereas the shorter chained fatty acids could be produced by microorganisms in the cave and the aquifer. This hypothesis seems to be supported by our LOP results.

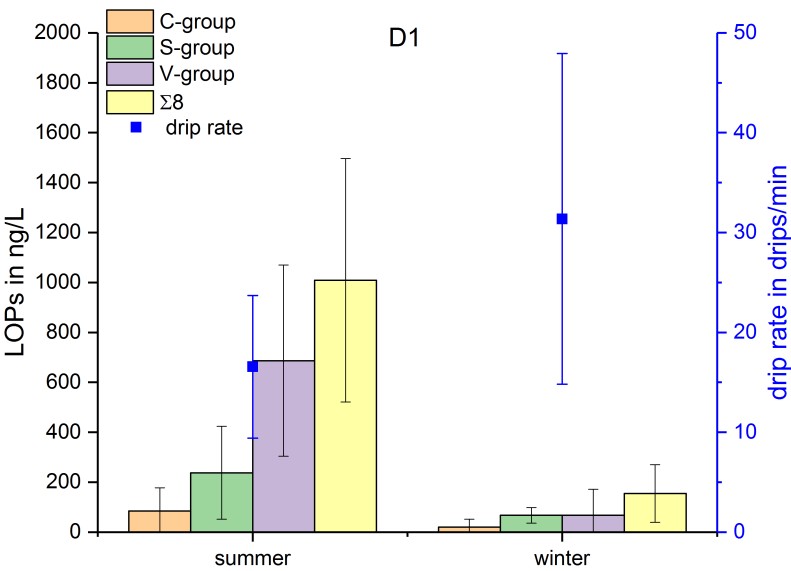

**Figure 8.** Comparison of Σ8 concentrations and drip rate in summer and winter.

In Fig. 9, the drip water Σ8 concentrations are compared with the drip rate and phosphate concentrations (Mischel et al., 2017). Phosphate shows high concentrations in autumn and winter and lower concentrations in spring and summer. Mischel et al. (2017) interpreted this pattern as resulting from organic material being flushed into the cave in autumn and winter. However, the low Σ8 concentrations in autumn and winter do not support this hypothesis. The difference in seasonal variations of Σ8 and phosphate could be explained either by different sources, as phosphate can also be leached from the bedrock, or by different transport mechanisms, such as transport in colloids or in solution. It is important to note that we are looking at the seasonal timescale here. On seasonal timescales, the variation of the phosphate concentration in the drip water is probably mainly influenced by direct contribution from the bedrock by different water flows, while the input of plant material from the reservoir of organic matter can be considered as relatively constant. On longer timescales with a resolution of decades to centuries, as recorded in the stalagmite, the link between phosphorous concentrations and the general activity of the vegetation may be stronger. This is because the release and mobilization of phosphorous from the bedrock through weathering is related to the productivity of the vegetation above the cave since phosphorous serves as a plant nutrient. In future studies, the analysis

of lignin and phosphate in drip water should be combined with total organic carbon measurement or fluorescence spectroscopy to better quantify the general input of organic matter.

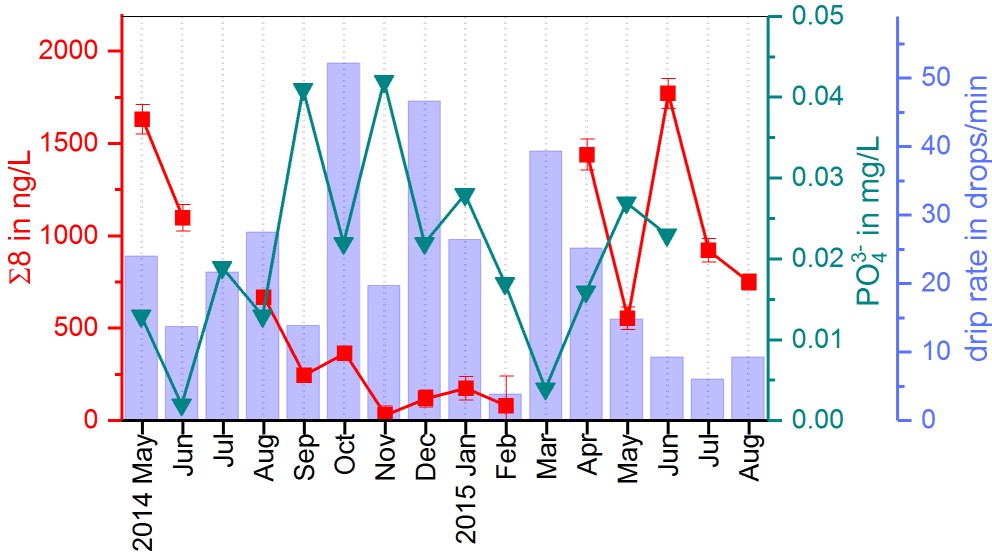

**Figure 9.** Comparison of Σ8 with $PO_4^{3-}$ concentrations and drip rate.

The record of the slow drip site, *D2*, shows no seasonal concentration pattern, but a high variability (Fig. 3). The pool water, *PW*, on the other hand, shows a smoothed out seasonality with low variance, with the exception of one month (April 2015) with exceptionally high concentrations. In Fig. 10, the averaged Σ8 concentrations of the fast drip site, *D1*, the slow drip site, *D2*, and the pool water, *PW*, are compared. On average, the slow drip site has a higher Σ8 concentration than the fast drip site, and the pool water has the lowest concentrations. This finding can be explained by dilution effects as discussed above, but also by different reservoirs feeding the individual drip sites.

The LOP concentrations in the drip water samples were generally quite low, and in some samples, individual analytes were below the limit of detection. Therefore, C/V and S/V ratios could not be calculated for every sample. The missing data points make the interpretation of C/V and S/V with respect to seasonal patterns rather difficult (see Fig. 4). If we consider only the common interpretation of C/V and S/V as vegetation source indicators, we would not expect to see a seasonal pattern here, as the overall vegetation mixture above the cave remained unchanged throughout one year and the turnover times of lignin degradation in soil are long. In addition, radiocarbon dating studies showed that the organic matter above caves can be a mix of recent material and material that is up to several hundred years old (Trumbore, 2000; Tegen and Dörr, 1996; Fohlmeister et al., 2011). On the other hand, considering fractionation processes as described by Hernes et al. (2007), changes in drip water C/V and S/V ratios may indicate changes in the transfer through the soil and karst system as well as changes in microbiological processes, such as degradation and interaction with soil organic matter, which are likely to change with temperature and moisture conditions.

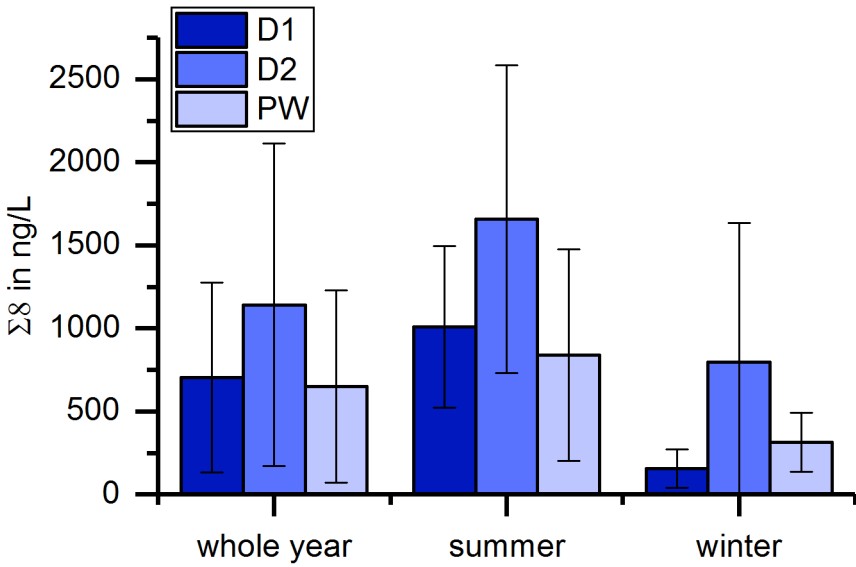

**Figure 10.** Comparison of average $\Sigma 8$ concentrations of the fast drip site, D1, the slow drip site, D2 and the poolwater, PW.

We also compared the C/V and S/V ratios with $\delta^{13}$C data and the drip rate (Fig. 11). From April to July 2015, there was a constant decrease in the drip rate and in the C/V and S/V ratios, whereas the $\delta^{13}$C values show a constant increase in the same time interval. The increase in $\delta^{13}$C values could be related to the decreasing drip rate and consequently a longer residence time of the drops in the cave air resulting in increased degassing (Hansen et al., 2017). An explanation for the decrease of C/V and S/V ratios could possibly be linked to the decreasing drip rate too. In times of reduced recharge of the aquifer, the transport of the lignin from the soil into the cave probably takes longer and involves more phase-changes than in times of higher recharge. According to Hernes et al. (2007) and Hernes et al. (2013), every phase-change that occurs to the lignin on its way from the plant litter to the deeper soil (and into the aquifer and the cave), for example from plant material to dissolved in water or from dissolved to adsorbed onto mineral surfaces, can lead to a fractionation and therefore to a change in the ratios of C/V, S/V as well as Acid/Aldehyde.

The C/V and S/V ratios of the drip water samples are lower than the ratios found in the stalagmite. This difference could also be caused by phase-change fractionation as the incorporation into the calcite of the growing speleothem represents a phase-change as well. To find out more about these effects, additional comparative studies of LOPs in speleothems and cave drip water as well as in situ experiments in an artificial cave (Hansen et al., 2017; Wiedner et al., 2008; Polag et al., 2010) would be beneficial.

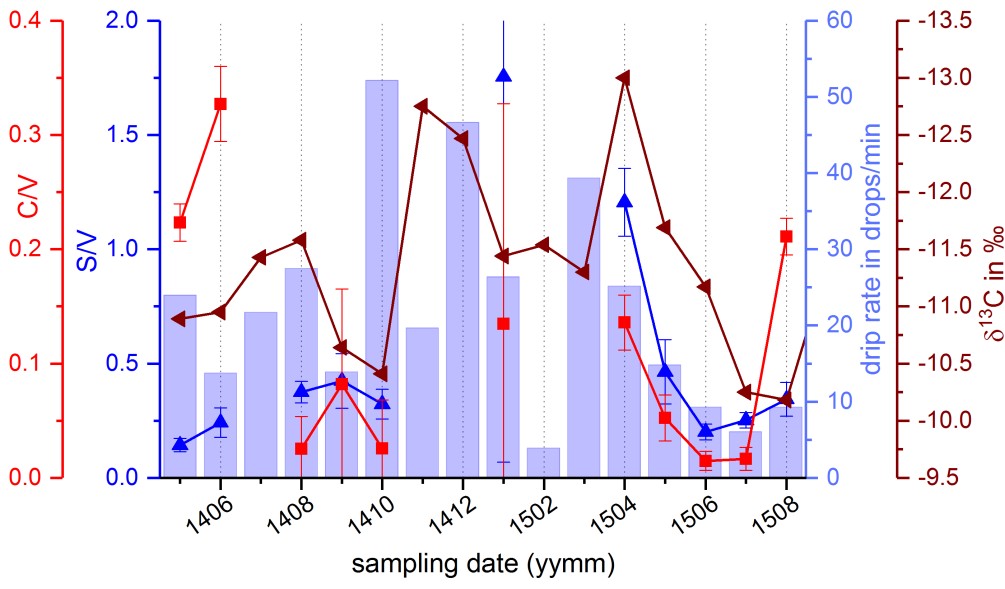

**Figure 11.** Comparison of C/V and S/V ratios with the drip rate and $\delta^{13}$C values in the fast drip site *D1*.

## 5 Conclusions and outlook

There are still many open questions concerning the interpretation of lignin oxidation products in speleothems and cave drip water. However, this first quantitative record of LOPs in a stalagmite and cave drip water shows that the quantification of LOPs is possible in both sample types, and that the signals show a significant variation over time on the centennial to millennial timescale in the stalagmite and a seasonal pattern in the drip water. The total LOP concentration, $\Sigma 8$, in the stalagmite is correlated to P, Ba and U concentrations interpreted as vegetation proxies (Mischel et al., 2017). The clear benefit of $\Sigma 8$ compared to these trace elements is that the sources of lignin are exclusively higher plants and not, for example, microorganisms or the host rock. Therefore, $\Sigma 8$ can complement or even help to better interpret potential vegetation proxies whose sources are less clear. On the other hand, lignin can also be affected by microbial processes, such as differential degradation or incorporation into larger aggregates of organic substances, and transport processes, such as the adsorption to mineral particles. These effects seem to have a larger influence on the lignin ratios C/V and S/V, which are usually used as vegetation source indicators. Therefore, more research is needed to unravel the different influences affecting lignin oxidation products in the karst system.

In future studies, more speleothem samples from different vegetation and climate zones should be analyzed to study the relation of vegetation types above the cave and LOP ratios found in the speleothems. To get more insight into possible fractionation processes occurring on the way from the soil to the cave, comparative studies of LOPs in soil, drip water and speleothems should be carried out and complemented with other methods, such as total organic carbon mesaurements, the calculation of C/N ratios or fluorescence spectroscopy. Annually laminated speleothems might be well suited as study objects because they

often possess high organic matter content and high seasonality. In addition to stalagmites, flowstones could be valuable sample material because they are often fed by water flows with higher discharge carrying larger amounts of organic material.

*Data availability.* We have provided all relevant data in the paper and the supplement to this study.

*Author contributions.* IH, DS, and TH designed the research; IH performed the research; IH, DS, and TH analyzed the data and all authors contributed to writing the paper

*Competing interests.* The authors declare that they have no conflict of interest.

*Disclaimer.* TEXT

5  *Acknowledgements.* We thank Simon Mischel for providing stalagmite and cave drip water samples from the Herbstlabyrinth Cave. This project has received funding from the European Union's Horizon 2020 research and innovation program under Marie Skłodowska-Curie grant agreement no. 691037. Denis Scholz acknowledges funding from the German Research Foundation (SCHO 1274/3-1, SCHO 1274/9-1 and SCHO 1274/11-1).

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
