# Peer review of "Lignin oxidation products as a potential proxy for vegetation and environmental changes in speleothems and cave drip water – a first record from the Herbstlabyrinth, central Germany"

_Climate of the Past, 2019_

## Referee Comment (RC1) · Anonymous Referee #1 · 14 Feb 2019

Heidke and colleagues present a quantitative record of lignin oxidation products (LOPs) from a Holocene stalagmite, together with the LOPs results from 16 months drip water monitoring in the same cave. By comparing with the previously published data, such as drip rate, d13C and elemental concentration, they argued that LOPs extracted from stalagmites had the potential to be acted as a highly specific vegetation proxy. With the merits of wide distribution and highly precise dating, it is worthy to evaluate the paleoe-cological potential of LOPs in speleothems, which can favor to decipher the relationship between vegetation and climate changes across multi timescales, and ultimately to aid to mitigate the potential influence of global warming on terrestrial ecosystems. However, the current manuscript seems like a report of their experimental data, and does

not encompass some general summaries to advocate its applications to other regions or different epochs. Thus, I suggest the authors to perform a major revision before the consideration for possible publication in the journal Climate of the Past.

The major concern is about the objectives and the new findings in this study. To be honest, reading of the current manuscript is not an easy task, particularly for audiences beyond the field of speleothems. In addition, it is not easy to extract the new findings in this study, and to be aware of how these findings can help us to improve the understanding of the processes that affect the transport of lignin into the cave and finally incorporated into the speleothems. In the introduction part, the authors did not clearly show their objectives. In the discussion part, they conducted many kinds of comparisons between the new and published data, and then present relatively complex explanations on the observed not simple relationships between proxies. Such a situation is clearly reflected in the abstract part. In the abstract, the authors mainly present detailed information about their results; however, summaries on the broad implications of the applications of this novel approach were nearly absent. Through the current text, evidence is not convinced to support the LOPs as a new, highly specific vegetation proxy in speleothems. In contrast, the audiences will be confused by the extremely complex pattern among LOPs proxies and the published ones from the same cave. In this way, the authors need to reframe the main text, and to focus how to constrain the major factors affecting the transport and preservation of LOPs in cave system on different timescales.

Another concern is on the integration of drip water data and the stalagmite data. In the current main text, these two datasets seem independent. These data were shown and discussed in different parts in the Results and Discussion parts separately. They need to explain why present these two types of data in the same paper, and to consider how to integrate these data. In fact, both data are useful for understanding the transport and preservation of LOPs in cave system. The data of drip water LOPs over the course of a hydrological year can yield useful information on seasonal timescale, though it is
cautious the influence of dwelling time in the epikarst zone. LOPs data from stalagmite facilitate to understand the processes on centennial to millennial timescales.

The final concern come from the interpretation of the seasonal variation of LOPs. D1 dripwater, Mg2+ and PO43- showed the opposite seasonal pattern compared with LOPs. The authors assumed the overlying reservoir was constant and highlighted the seasonal signal of LOPs was mainly controlled by hydrological effects, e.g. LOPs are be diluted in winter when recharge is higher; however, according to this mode, the seasonal variation of Mg2+ and PO43- in the dripping water will be similar to the pattern of Lops, other than the observed results (lower value in dryer summer and higher value in wetter winter)

Some minor comments: L8-9 in abstract, "The LOP ratios C/V and S/V, which are usually used to differentiate between angiosperm and gymnosperm and woody and non-woody lignin sources" should be "The LOP ratios S/V and C/V,....". L15 in P1: 500.000 years? If this is right, why keep three decimals? By the way, The 230 Th-U dating range has been expanded to 640.000 years (Cheng et al., 2016). L8 in P2: italic 'n'. P4: suggest to delete the only subsection '2.3.1'. Fig. 3. Annotation is required for the grey lines in the uppermost panel. In addition, please add the title of the y-axis for this panel. Fig. 9. Correct the title of the y-axis and add the unit for the oxygen isotope data.

CPD

---

## Referee Comment (RC2) · Yves Perrette (Referee) · 4 Mar 2019

Lignin oxidation products as a vegetation proxy in stalagmite and drip water samples from the Herbstlabyrinth, Germany

This good scientific work coupled the study of lignin products in dripwater and in stalagmite to discuss the interest of these biomarkers as vegetation proxy. It is well written and illustrations are good and readable. But, results don't support that so accurate title. This work demonstrates the interest of lignin metabolites as Âń global environmental" proxy including soil activity, climate and vegetation which have to be unraveled.

[Figure]

I recommend moderates reviews

My general concern is the need of a more detailed discussion of the transfer of these biomarkers through karst system. This work discusses the lignin transfer without taking into account their interaction with organic substances in the soil, during the transfer and in the stalagmite. Authors should include some literature and discussion about that link (see Lehman et al 2014 for instance).

A second concern more specific is the lack of a discussion of the Wynn et Brooks (20014) paper which give an accurate discussion about methodology for biomarker analyses in speleothem.

More detailed comments are done on the PDF.

Please also note the supplement to this comment:
https://www.clim-past-discuss.net/cp-2019-5/cp-2019-5-RC2-supplement.pdf

—————————————————

[Figure]

**Supplement:**

[revised manuscript text omitted]

---

## Short Comment (SC1) · 11 Mar 2019

Dear Yves Perrette,

thank you for your review, I will answer in detail later, but I have got one short question: I am not sure which reference you mean by Lehman et al. 2014, do you mean Lehmann, J., Kleber, M. (2015). The contentious nature of soil organic matter. Nature. 528:60-68?

Best regards,

Inken Heidke

---

## Author Comment (AC1) · 4 Apr 2019

We thank the two reviewers for carefully evaluating our manuscript. Our point-by-point reply directly follows the referee comments (blue) and appears in black after each comment. When we refer to short text passages of the manuscript, we use quotation marks, new or revised text segments are printed in green. As we performed a major revision, the revised manuscript is attached at the end of the point-by-point reply.

5 **Response to referee #1, anonymous**

Referee Comment 1:
the current manuscript seems like a report of their experimental data, and does not encompass some general summaries to advocate its applications to other regions or different epochs.
We agree with the referee and have taken two measures to change this: We have stated our objectives more clearly in the
10 introduction, see reply to comment 2, and we have included more general summaries and advices about the application of our method. Here are some examples from different text passages.
In the introduction:
"The aim of this study is to present first results on the application of the lignin analysis method developed by Heidke et al. (2018) to speleothem and cave drip water samples to generally evaluate the potential of LOPs as a proxy for vegetational and
15 environmental changes in speleothem archives. We present the first quantitative record of lignin oxidation products from a Holocene speleothem from the Herbstlabyrinth in central Germany. Our objectives were (i) to investigate, if the sensitivity of our method is high enough to detect and quantify LOPs with a sufficient temporal resolution to reveal centennial to millennial climate changes, and (ii), if and how the LOP signal in speleothems varies with climatic and vegetational changes and how it compares with other, established proxy signals, such as trace elements and stable isotopes."
20 In the discussion part:
"To further study the potential of LOPs as a vegetation proxy in the future, it would be beneficial to use speleothem samples from an epoch with more pronounced vegetation changes."
"For future experiments, it might be more useful to apply long-term sampling devices, such as in-situ preconcentration with solid-phase extraction cartridges, instead of whole-water sampling once per month. This would allow quantification of the
25 input of lignin per month instead of the concentration per liter drip water."
"In future studies, the analysis of lignin and phosphate in drip water should be combined with total organic carbon measurement or fluorescence spectroscopy to better quantify the general input of organic matter."
And in the conclusion:
"In future studies, more speleothem samples from different vegetation and climate zones should be analyzed to study the
30 relation of vegetation types above the cave and LOP ratios found in the speleothems. To get more insight into possible fractionation processes occurring on the way from the soil to the cave, comparative studies of LOPs in soil, drip water and speleothems should be carried out and complemented with other methods, such as total organic carbon mesasurements, the calculation of C/N ratios or fluorescence spectroscopy. Annually laminated speleothems might be well suited as study objects because they often possess high organic matter content and high seasonality. In addition to stalagmites, flowstones could be valuable sample
35 material because they are often fed by water flows with higher discharge carrying larger amounts of organic material."

Referee Comment 2:
The major concern is about the objectives and the new findings in this study. [...] In the introduction part, the authors did not clearly show their objectives.
We have rewritten parts of the introduction to make our objectives clear. We now write in the last two paragraphs of the intro-
40 duction:
"It is still a subject of research to what extent and how these ratios are affected by factors such as transport, microbial transformation, soil characteristics and land use as well as the interaction with mineral surfaces (Hernes et al., 2007; Thevenot et al., 2010; Hernes et al., 2013; Jex et al., 2014). A clarification of all these aspects is beyond the scope of this study. The aim of this study is to present first results on the application of the lignin analysis method developed by Heidke et al. (2018) to speleothem

and cave drip water samples to generally evaluate the potential of LOPs as a proxy for vegetational and environmental changes in speleothem archives. We present the first quantitative record of lignin oxidation products from a Holocene speleothem from the Herbstlabyrinth in central Germany. Our objectives were (i) to investigate, if the sensitivity of our method is high enough to detect and quantify LOPs with a sufficient temporal resolution to reveal centennial to millennial climate changes, and (ii), if and how the LOP signal in speleothems varies with climatic and vegetational changes and how it compares with other, established proxy signals, such as trace elements and stable isotopes. Therefore, we compared our LOP results with trace element and stable isotope records from the same sample (Mischel et al., 2017). In addition, we aimed to improve our knowledge on how lignin is transported into the cave, which is a key question in understanding LOPs as an environmental proxy in speleothem archives. Therefore, we investigated seasonal variations of LOP concentrations in drip water from the same cave, sampled monthly over a period of 16 months in the framework of a cave monitoring program.

We chose the Herbstlabyrinth Cave for our first quantitative LOP analysis because of an ongoing cave monitoring program, and the specific Holocene stalagmite was chosen because it was already well characterized by stable isotope and trace element analysis, the general vegetation and climate changes in the Holocene in central Germany are relatively well known (e. g. Litt et al., 2009), and last but not least, the fast growth rate of the stalagmite offered sufficient sample material to test and apply our method."

Referee Comment 3:
To be honest, reading of the current manuscript is not an easy task, particularly for audiences beyond the field of speleothems. In addition, it is not easy to extract the new findings in this study, and to be aware of how these findings can help us to improve the understanding of the processes that affect the transport of lignin into the cave and finally incorporated into the speleothems. [...] In the discussion part, they conducted many kinds of comparisons between the new and published data, and then present relatively complex explanations on the observed not simple relationships between proxies. Such a situation is clearly reflected in the abstract part. In the abstract, the authors mainly present detailed information about their results; however, summaries on the broad implications of the applications of this novel approach were nearly absent.

We have tried to improve the readability of the manuscript and especially of the discussion part, see revised manuscript at the end of this document. The basic structure of the discussion has been revised. The discussion part now starts with the discussion of the stalagmite samples, and here we begin with the discussion of the principal component analysis to provide an overview of the correlations between the LOPs and the trace element and stable isotope data. To improve the reading flow, the individual correlation coefficients are not mentioned in the text anymore, but are presented in tables in the supplementary information. After this general overview, we discuss two selected peaks in the records in greater detail with respect to vegetation and climate changes. In the end of the stalagmite section, we compare our findings with pollen-based climate reconstructions.

The discussion of the drip water samples starts with a general interpretation of the seasonal pattern of the fast drip site, *D1*, followed by a comparison with fatty acid biomarkers (Bosle et al., 2014) as well as phosphate concentrations and the drip rate (Mischel et al., 2017). The comparison with the Mg concentrations was omitted, because it does not contribute to the discussion of lignin in drip water. This is followed by the comparison of *D1*, *D2* and *PW*, and the section ends with the discussion of the C/V and S/V ratios.

We also revised the abstract to make our objectives and our findings more clear:

"Here we present the first quantitative speleothem record of lignin oxidation products (LOPs), which has been determined in a Holocene stalagmite from the Herbstlabyrinth Cave in central Germany. In addition, we present LOP results from 16 months of drip water monitoring. Lignin is only produced by vascular plants and therefore has potential as an unambiguous vegetation proxy and to complement other vegetation and climate proxies in speleothems. We compare our results with stable isotope and trace element data from the same sample. In the stalagmite, LOP concentrations show a similar behaviour as P, Ba and U concentrations, which have previously been interpreted as vegetation proxies. The LOP ratios S/V and C/V, which are usually used to differentiate between angiosperm and gymnosperm and woody and non-woody vegetation, show complex patterns suggesting additional influencing factors, such as transport and microbiological effects. The drip water from a fast drip site shows a seasonal pattern of LOPs with low LOP concentrations in winter and higher LOP concentrations in summer. These results indicate the potential of LOPs as a new proxy for vegetational and environmental changes in speleothems, but also

demonstrate the complexity and the current limitations of our understanding of the transport of lignin from the soil into the cave and the speleothems."

Referee Comment 4:

Through the current text, evidence is not convinced to support the LOPs as a new, highly specific vegetation proxy in speleothems. In contrast, the audiences will be confused by the extremely complex pattern among LOPs proxies and the published ones from the same cave. In this way, the authors need to reframe the main text, and to focus how to constrain the major factors affecting the transport and preservation of LOPs in cave system on different timescales.

The main text was reframed. We revised the aims and objectives in the introduction (see reply to comment 2) to make clear that we want to "present first results on the application of the lignin analysis method developed by (Heidke et al., 2018) to speleothem and cave drip water samples to generally evaluate the potential of LOPs as a proxy for vegetational and environmental changes in speleothem archives". We revised the discussion (see reply to comment 3), now focusing on the main aspects only. We now try to make clear at many points in the text that the processes affecting the transport and preservation of LOPs in cave systems are complex. However, a clarification of all these aspects is beyond the scope of this study. Consequently, we adjusted the conclusion according to the objectives of the study, now stating that There are still many open questions concerning the interpretation of lignin oxidation products in speleothems and cave drip water. However, this first quantitative record of LOPs in a stalagmite and cave drip water shows that the quantification of LOPs is possible in both sample types, and that the signals show a significant variation over time on the centennial to millennial timescale in the stalagmite and a seasonal pattern in the drip water.", and also that "lignin can also be affected by microbial processes, such as differential degradation or incorporation into larger aggregates of organic substances, and transport processes, such as the adsorption to mineral particles. These effects seem to have a larger influence on the lignin ratios C/V and S/V, which are usually used as vegetation source indicators. Therefore, more research is needed to unravel the different influences affecting lignin oxidation products in the karst system." For the whole text, see the revised manuscript at the end of this document.

Referee Comment 5:

Another concern is on the integration of drip water data and the stalagmite data. In the current main text, these two datasets seem independent. These data were shown and discussed in different parts in the Results and Discussion parts separately. They need to explain why present these two types of data in the same paper, and to consider how to integrate these data. In fact, both data are useful for understanding the transport and preservation of LOPs in cave system. The data of drip water LOPs over the course of a hydrological year can yield useful information on seasonal timescale, though it is cautious the inuence of dwelling time in the epikarst zone. LOPs data from stalagmite facilitate to understand the processes on centennial to millennial timescales.

We agree with the reviewer that the connection between both data sets was not sufficiently explained in the previous version of the manuscript. We still keep the results and the discussion of drip water and stalagmite data in different sections because we think it provides orientation for the reader, but we have added some explaining sentences. In the introduction, we now write:

"In addition, we aimed to improve our knowledge on how lignin is transported into the cave, which is a key question in understanding LOPs as an environmental proxy in speleothem archives. Therefore, we investigated seasonal variations of LOP concentrations in drip water from the same cave, sampled monthly over a period of 16 months in the framework of a cave monitoring program."

In the beginning of the discussion section on the drip water samples (which is now situated after the discussion of the stalagmite samples), we now write:

"To gain a better understanding of the transport mechanisms and preservation of LOPs in the cave system, we complemented our first stalagmite LOP record with the analysis of LOPs in monthly sampled drip water (Fig. 5). The study of LOP concentrations over the course of a hydrological year can yield useful information on transport processes and fluxes on a seasonal timescale, provided that the influence of a prolonged dwelling time in the epikarst zone is taken into account. By comparison

of modeled and measured $\delta^{18}O$ values, Mischel et al. (2015) have calculated that the drip water of the Herbstlabyrinth is fed by a large reservoir in the karst aquifer, where the water has a residence time of approx. 10 months and consists of rain water mixed over a period of ca. 12 months. Therefore, seasonal variations are strongly attenuated, but should not be shifted much."
We also tried to emphasize that the timescales are different for drip water and speleothem analysis and that the signals therefore can be influenced by different processes. Here is an example from the drip water discussion:
"The difference in seasonal variations of $\Sigma 8$ and phosphate could be explained either by different sources, as phosphate can also be leached from the hostrock, or by different transport mechanisms, such as transport in colloids or in solution. It is important to note that we are looking at the seasonal timescale here. On longer timescales, as recorded in the stalagmite, the release and mobilization of phosphorous from the hostrock can be related to the productivity of the vegetation above the cave since phosphorous serves as a plant nutrient."

Referee Comment 6:
The final concern come from the interpretation of the seasonal variation of LOPs. D1 dripwater, Mg2+ and PO43- showed the opposite seasonal pattern compared with LOPs. The authors assumed the overlying reservoir was constant and highlighted the seasonal signal of LOPs was mainly controlled by hydrological effects, e.g. LOPs are be diluted in winter when recharge is higher; however, according to this mode, the seasonal variation of Mg2+ and PO43- in the dripping water will be similar to the pattern of Lops, other than the observed results (lower value in dryer summer and higher value in wetter winter)
We understand your point and have revised the discussion of the seasonal variations in the drip water. After studying the original data in Mischel et al. (2017) again, we have omitted the Mg record because it does not contribute to the discussion of lignin in drip water. Instead, we focus on the comparison of $\Sigma 8$ and phosphate only and now write:
"In Fig. 11, the drip water $\Sigma 8$ concentrations are compared with the drip rate and phosphate concentrations (Mischel et al., 2017). Phosphate shows high concentrations in autumn and winter and lower concentrations in spring and summer. Mischel et al. (2017) interpreted this pattern as resulting from organic material being flushed into the cave in autumn and winter. However, the low $\Sigma 8$ concentrations in autumn and winter do not support this hypothesis. The difference in seasonal variations of $\Sigma 8$ and phosphate could be explained either by different sources, as phosphate can also be leached from the hostrock, or by different transport mechanisms, such as transport in colloids or in solution. It is important to note that we are looking at the seasonal timescale here. On longer timescales, as recorded in the stalagmite, the release and mobilization of phosphorous from the hostrock can be related to the productivity of the vegetation above the cave since phosphorous serves as a plant nutrient. In future studies, the analysis of lignin and phosphate in drip water should be combined with total organic carbon measurement or fluorescence spectroscopy to better quantify the general input of organic matter."

Referee Comment 7, minor coments:

1. L8-9 in abstract, "The LOP ratios C/V and S/V, which are usually used to differentiate between angiosperm and gymnosperm and woody and non-woody lignin sources" should be "The LOP ratios S/V and C/V,. . ...".
   Has been corrected.

2. L15 in P1: 500.000 years? If this is right, why keep three decimals? By the way, The 230 Th-U dating range has been expanded to 640.000 years (Cheng et al., 2016).
   Thank you for spotting this typo and adding the reference. We have corrected the dating range and now write "Speleothems are valuable climate archives, because they can grow continuously for thousands of years and can be dated accurately 640,000 years back in time using the $^{230}$Th-U method (Cheng et al., 2016; Richards and Dorale, 2003; Scholz and Hoffmann, 2008)."

3. L8 in P2: italic 'n'.
   Has been corrected.

4. P4: suggest to delete the only subsection '2.3.1'.
   The subsection has been deleted and the paragraph has been restructured, now starting with "The analytical method for the analysis of LOPs is described in detail in Heidke et al. (2018). In brief, the stalagmite samples for LOP analysis were cleaned with organic solvents,..."

5. Fig. 3. Annotation is required for the grey lines in the uppermost panel. In addition, please add the title of the y-axis for this panel.
   The grey lines (connected error bars) in Fig. 3 and Fig. 4 have been replaced by normal error bars to avoid confusion.

6. Fig. 9. Correct the title of the y-axis and add the unit for the oxygen isotope data.
   The titles of the Y-axis for the $\delta^{13}$C and $\delta^{18}$O data have been corrected, thank you for spotting these mistakes.

**Response to referee #2, Yves Perette**

Referee Comment 1:
Results don't support that so accurate title. This work demonstrates the interest of lignin metabolites as a "global environmental" proxy including soil activity, climate and vegetation which have to be unraveled.
We agree with the reviewer and have changed the title to
"First quantitative record of lignin oxidation products as a potential proxy for vegetation and environmental changes in stalagmite and drip water samples"

Referee Comment 2:
My general concern is the need of a more detailed discussion of the transfer of these biomarkers through karst system. This work discusses the lignin transfer without taking into account their interaction with organic substances in the soil, during the transfer and in the stalagmite. Authors should include some literature and discussion about that link (see Lehman et al 2014 for instance).
We have included a more detailed discussion of the lignin transfer at several points, see individual replies to the "Referee Comments made on the pdf". In addition, we have adjusted our aims and objectives to make clear that an in-depth discussion of all aspects is beyond the scope of the manuscript. In the following are some examples from the revised text, the revised manuscript is attached at the end of this document.
In the introduction:
"It is still a subject of research to what extent and how these ratios are affected by factors such as transport, microbial transformation, soil characteristics and land use as well as the interaction with mineral surfaces (Hernes et al., 2007; Thevenot et al., 2010; Hernes et al., 2013; Jex et al., 2014). A clarification of all these aspects is beyond the scope of this study. The aim of this study is to present first results on the application of the lignin analysis method developed by Heidke et al. (2018) to speleothem and cave drip water samples to generally evaluate the potential of LOPs as a proxy for vegetational and environmental changes in speleothem archives."
In the discussion part:
"The fact that $\Sigma$8 and C/V and S/V appear in both PC1 and PC2 indicates that the $\Sigma$8 concentration and especially the C/V and S/V ratios are not only influenced by the abundance and type of vegetation, but also by hydrological and soil microbiological effects, such as the transport of organic matter through the soil and the karst system."
"As the transport of lignin is likely to involve processes such as binding to particles or coiling with humic substances, many different factors may play a role here."
"On the other hand, considering fractionation processes as described by Hernes et al. (2007), changes in drip water C/V and

S/V ratios may indicate changes in the transfer through the soil and karst system as well as changes in microbiological processes, such as degradation and interaction with soil organic matter, which are likely to change with temperature and moisture conditions."

In the conclusion:

5 "On the other hand, lignin can also be affected by microbial processes, such as differential degradation or incorporation into larger aggregates of organic substances, and transport processes, such as the adsorption to mineral particles. These effects seem to have a larger influence on the lignin ratios C/V and S/V, which are usually used as vegetation source indicators. Therefore, more research is needed to unravel the different influences affecting lignin oxidation products in the karst system."

Referee Comment 3:

10 A second concern more specic is the lack of a discussion of the Wynn et Brooks (2014) paper which give an accurate discussion about methodology for biomarker analyses in speleothem.

We agree with the referee and have included the reference to the Wynn and Brocks (2014) paper in the introduction and the analytical methods section, see replies to the "Referee Comments made on the pdf" number 3 and 7. We did not conduct an experiment exactly like the one suggested by Wynn and Brocks to examine the spacial distribution of the lignin and LOP

15 concentrations in the speleothem samples. However, we tried to reduce the risk of laboratory contamination by edging away the outer layer of calcite of all our samples. In addition, our analytes are very specifically produced by plants and are not as prone to contamination as the ubiquitous lipids (e. g. sterol or fatty acids) analyzed by Wynn and Brocks. Apart from that, we have already discussed the issues of laboratory contamination and blank values in detail in our method paper (Heidke et al., 2018).

20 Referee Comments made on the pdf:

1. Title, "as a vegetation proxy": as a proxy of environmental changes in, environment as a mixed of vegetation, soils activity and climate

   We changed the title to "First quantitative record of lignin oxidation products as a potential proxy for vegetation and

25 environmental changes in speleothems and cave drip water"

2. P1, L17, "mechanical disturbance": Need to be temperered .. many speleothems can be weathered and or eroded in cave due to sediment infills water

   We changed the sentence to "Furthermore, the cave provides a preservative environment that protects the recorded chemical proxy signals against outer influences such as light, abrupt changes in temperature and, under ideal conditions,

30 also mechanical disturbance."

3. P2, L6: the pbm of calcite contamination must be mentioned here as a thread on such analysis (Wynn and brooks 2014)

   We agree with the referee and now write:

   "Lipid biomarkers, such as fatty acids (**?**) and especially long chain *n*-alkanes from plant leave waxes, have been used as vegetation proxies, and there have been approaches to use the chain length distribution of *n*-alkanes to distinguish

35 between the input of grasses and woody plants (Xie, 2003; Blyth et al., 2007, 2011). However, there are uncertainties about the validity of chain length distributions to distinguish between different plant groups (Bush and McInerney, 2013; Blyth et al., 2016). In addition, lipid biomarkers are especially prone to laboratory contamination, which poses a general problem for biomarker analysis (Wynn and Brocks, 2014). Therefore, a more specific plant biomarker is needed that is less prone to contamination."

40 4. P2, L25–33: One of the main concern is what is the link between humic substances (lignin can be included in the precursors of HS), it may not change the tie with sources but the transfer from soils to underground waters and then speleothems

Of course, lignins are part of the soil organic matter and can undergo various biological, physical and chemical transformations. We agree with the referee that these transformations can influence the LOP concentrations and C/V and S/V ratios measured in the speleothem samples and therefore their interpretation as proxies for vegetation and environmental changes. However, it is not the aim of our publication to discuss all these transformations in detail, but to present first results on the analysis of LOPs in speleothems. Therefore, we have reformulated our aims and objectives in the introduction, see our response to referee #1, comment 2. Further systematic studies about the links between LOPs in plant, soil, drip water and speleothem samples are in progress in our group.

5. P2, L25–33: Ok for the signature of the LOP Sources, but what about the transfer through the karst system... do all these lignine have the same physical specificties (adsorption, reactivity, biodegradability)

This is an interesting question, and we do not know exactly how lignin is transported yet. As stated above, we have adjusted the objectives of this paper, and further systematic studies are in progress.

6. P4, L7–8: for comparison, you should had the standard deviation around to have an idea of the dispersions. of high resolutions data.

The standard deviations are given as error bars in the figures, as stated in Line 9. For better readability, we have restructured the sentence and now write:

"To compare the LOP results with stalagmite data of stable isotopes and trace elements, which have a much higher resolution of 2 mm per sample, a mean value of the higher-resolution data according to the sample size of the LOP samples was calculated, and the standard deviation of these mean values was used as error-bars."

7. P4, L15–16: wynn et brooks...

We have included the reference:
"In brief, the stalagmite samples for LOP analysis were cleaned with organic solvents, edged on the outside with diluted HCl to prevent the influence of potential laboratory contamination (Wynn and Brocks, 2014), and finally dissolved in ultra pure 30% HCl."

8. P13, L5–7: this pattern may only mean that there is a shift in the ratio due to differential level of incorporation in humic or differential transfers through the karst system (sources of soils depending of the soil water saturation state) You should include that possiblity in your discusssion... as in the discussion of the speleothem signature

The discussion of the transfer through the karst system and the incorporation in humic substances had been included at several points in the text. The discussion of the drip water C/V and S/V ratios now is as follows:

"The LOP concentrations in the drip water samples were generally quite low, and in some samples, individual analytes were below the limit of detection. Therefore, C/V and S/V ratios could not be calculated for every sample. The missing data points make the interpretation of C/V and S/V with respect to seasonal patterns rather difficult (see Fig. 6). If we consider only the common interpretation of C/V and S/V as vegetation source indicators, we would not expect to see a seasonal pattern here, as the overall vegetation mixture above the cave remained unchanged throughout one year and the turnover times of lignin degradation in soil are long. In addition, radiocarbon dating studies showed that the organic matter above caves can be a mix of recent material and material that is up to several hundred years old (Trumbore, 2000; Tegen and Dörr, 1996; Fohlmeister et al., 2011). On the other hand, considering fractionation processes as described by Hernes et al. (2007), changes in drip water C/V and S/V ratios may indicate changes in the transfer through the soil and karst system as well as changes in microbiological processes, such as degradation and interaction with soil organic matter, which are likely to change with temperature and moisture conditions.

We also compared the C/V and S/V ratios with $\delta^{13}$C data and the drip rate (Fig. 13). From April to July 2015, there was a constant decrease in the drip rate and in the C/V and S/V ratios, whereas the $\delta^{13}$C values show a constant increase in the same time interval. The increase in $\delta^{13}$C values could be related to the decreasing drip rate and consequently a longer residence time of the drops in the cave air resulting in increased degassing (Hansen et al., 2017). An explanation for the decrease of C/V and S/V ratios could possibly be linked to the decreasing drip rate too. In times of reduced recharge of the aquifer, the transport of the lignin from the soil into the cave probably takes longer and involves more phase-changes than in times of higher recharge. According to Hernes et al. (2007) and Hernes et al. (2013), every phase-change that

occurs to the lignin on its way from the plant litter to the deeper soil (and into the aquifer and the cave), for example from plant material to dissolved in water or from dissolved to adsorbed onto mineral surfaces, can lead to a fractionation and therefore to a change in the ratios of C/V, S/V as well as Acid/Aldehyde."

9. P16, L3: what sort of correlation, Pearson, ? you should try Spearman which don't assume a linear trend between both variables but an ordinal correlation only.

   Yes, we have used Pearson correlation. Now, we have calculated both Pearson and Spearman correlations, but the differences are not very large. We tried to improve the readability of the main text by omitting all correlation coefficients in the text and have put them all in the supplementary information.

10. P16, L19–21: can you quantify global organic matter in the stalagmite ? it should enlight all that discussion (quiers et al 2015)

    Thank you for this suggestion. We cannot quantify global organic matter in this stalagmite anymore, but we have noted this for future analyses. We now write:

    "Therefore, it is also possible that a change in the flow path of the drip water feeding the stalagmite was responsible for the faster growth of the speleothem and a higher input of organic matter. To further investigate this kind of questions, it might be useful in the future to combine LOP analysis with total organic carbon measurements and fluorescence spectroscopy to gain more information on the amount and composition of organic matter in the speleothem."

11. P16, L33–34: It is difficult to consider the transfer of lignin (and others organics and minerals compounds) free in solution. Complexation or coiling in humic substance must be considered

    We did not consider the transfer of lignin free in solution. To make this more clear, we now write:

    "However, another possible interpretation, which would also be in accordance with the other proxies, would be that during drier conditions, the lignin was transported more slowly through the aquifer and into the cave and therefore had more time to undergo phase-changes on the way, such as adsorption to mineral surfaces. Such phase-changes can also lead to a change in the C/V and S/V ratios (Hernes et al., 2007). As the transport of lignin is likely to involve processes such as binding to particles or coiling with humic substances, many different factors may play a role here."

12. P17, Table 1: pearson ?

    Yes, Pearson. See reply to comment 9.

13. P18, L16–18:

    (a) from a methodological point of view, it may be more evident for me as reader to begin by the pca analyses

        Thank you for this suggestion. We now start the discussion of the stalagmite data with the PCA, before we come to a more detailed discussion of the records. The section now starts with:

        "First of all, we can state that LOPs were detectable in all stalagmite samples and above the quantification limit. Moreover, the signals show a variation over time on the centennial to millennial scale. In Fig. 7, the lignin parameters $\Sigma 8$, C/V and S/V of the stalagmite samples are compared with several trace elements, stable isotopes and the growth rate of the stalagmite.

        To get an overview of the correlations between the LOPs and the trace element and stable isotope data, we performed a principal component analysis (PCA).[1] As the three growth phases of the stalagmite show a different behaviour in several proxies and the growth rate (Mischel et al., 2017), the PCA and the correlation coefficients were calculated separately for each growth phase. As the youngest part consists of only five LOP samples, we focus the discussion on the middle and older part. (The contribution coefficients of the PCA are shown in Table 7 in the SI, the eigenvalues and percentages of variance in Table 8 in the SI, and all correlation coefficients of both Pearson's linear correlation and Spearman's rank correlation are shown in Tables 1 to 6 in the SI.)

        The PCA for the middle part is shown in Fig. 8. In the middle part, principal component 1 (PC1) explains...

        [...] The PCA for the older part is shown in Fig. 9. In the older part of the stalagmite, PC1 explains..."
* * *
[1]The PCA is based on Pearson's linear correlation. In contrast to Mischel et al. (2017), we did not detrend the records before calculating the correlations.

[...] Taking a closer look at the records in Fig. 7, there is a peak in the growth rate at the beginning of the middle part (around 8.0–7.5 ka BP) that is accompanied by a peak in Σ8 and also in P, Ba, and U concentrations. A higher growth rate can be caused..."

(b) your first component validate the idea (which seems the more grounded) that quality changes are more associated to process (in the soil or during the transfer) than to a vegetation change

As we have now calculated the PCA separately for the middle part and the older part of the stalagmite, the principal components have changed a bit (Fig. 1 and Fig. 2), but we have included the idea of the reviewer's comment. The discussion of the PCA is now as follows:

The PCA for the middle part is shown in Fig. 1. In the middle part, principal component 1 (PC1) explains 45.0% of the overall variance and consists mainly of P, Ba, U and Σ8 with positive contributions and C/V, S/V and $\delta^{13}$C with negative contributions. Mischel et al. (2017) interpreted P, Ba and U in the Herbstlabyrinth as vegetation proxies, with higher concentrations of these elements indicating a more productive vegetation, coinciding with wetter climate conditions. Since lignin has its source unambiguously in the vegetation, the correlation of Σ8 with P, Ba and U supports this interpretation. $\delta^{13}$C values were interpreted as being at least partially influenced by soil $pCO_2$ and soil thickness and thus indirectly affected by vegetation changes (Mischel et al., 2017). PC2 explains 23.3% of the variation and consists mainly of $\delta^{18}$O, Mg and the growth rate on the positive side (with smaller contributions of Σ8) and, with smaller coefficients, U, C/V and S/V on the negative side. The fact that Σ8 and C/V and S/V appear in both PC1 and PC2 indicates that the Σ8 concentration and especially the C/V and S/V ratios are not only influenced by the abundance and type of vegetation, but also by hydrological and soil microbiological effects, such as the transport of organic matter through the soil and the karst system.

The PCA for the older part is shown in Fig. 2. In the older part of the stalagmite, PC1 explains 51.1% and

[Figure]

**Figure 1.** Principal component analysis for the middle part of the stalagmite.

consists mainly of C/V, S/V, U and Ba on the positive side and Sr and Mg on the negative side. It can be inferred from the scores of the individual samples that the influence of Sr and Mg is mainly dominant in the older part

[Figure]

**Figure 2.** Principal component analysis for the older part of the stalagmite.

of the stalagmite. This long-term decrease of Sr and Mg (see Fig. 7) was interpreted by Mischel et al. (2017) as the result of a thin loess cover, deposited during the last Glacial, being progressively leached at the beginning of the Holocene. PC2 explains 20.4% and consists mainly of positive contributions of the growth rate and negative contributions of $\delta^{13}$C and $\delta^{18}$O. $\Sigma8$ only appears in the third principal component (explaining 11.4%), with a high positive contribution, together with positive contributions of P (see table **??** in the SI). This suggests that in the older part of the stalagmite, the influence of vegetation changes on the speleothem signals only plays a subordinate role.

14. P18, L21: that could be measured...
    See reply to comment 10 and the new discussion of the PCA in the reply to comment 13 b).

15. P18, L23: no, it may be the opposite, it could confirm that CV and SV ratio are some soil activity indicator rather than vegetation proxies
    The discussion was revised. We do not claim anymore to confirm P, Ba and U as vegetation proxies, but we now write in the discussion of the PCA: "Since lignin has its source unambiguously in the vegetation, the correlation of $\Sigma8$ with P, Ba and U supports this interpretation." And in the discussion of the C/V and S/V ratios: "The fact that $\Sigma8$ and C/V and S/V appear in both PC1 and PC2 indicates that the $\Sigma8$ concentration and especially the C/V and S/V ratios are not only influenced by the abundance and type of vegetation, but also by hydrological and soil microbiological effects, such as the transport of organic matter through the soil and the karst system."

16. P18, L26–27: for me, this conclusion is overinterpretated
    We revised our conclusion and now write:
    "There are still many open questions concerning the interpretation of lignin oxidation products in speleothems and cave drip water. However, this first quantitative record of LOPs in a stalagmite and cave drip water shows that the quantification of LOPs is possible in both sample types, and that the signals show a significant variation over time on the

centennial to millennial timescale in the stalagmite and a seasonal pattern in the drip water. The total LOP concentration, $\Sigma 8$, in the stalagmite is correlated to P, Ba and U concentrations interpreted as vegetation proxies (Mischel et al., 2017). The clear benefit of $\Sigma 8$ compared to these trace elements is that the sources of lignin are exclusively higher plants and not, for example, microorganisms or the host rock. Therefore, $\Sigma 8$ can complement or even help to better interpret potential vegetation proxies whose sources are less clear. On the other hand, lignin can also be affected by microbial processes, such as differential degradation or incorporation into larger aggregates of organic substances, and transport processes, such as the adsorption to mineral particles. These effects seem to have a larger influence on the lignin ratios C/V and S/V, which are usually used as vegetation source indicators. Therefore, more research is needed to unravel the different influences affecting lignin oxidation products in the karst system.

In future studies, more speleothem samples from different vegetation and climate zones should be analyzed to study the relation of vegetation types above the cave and LOP ratios found in the speleothems. To get more insight into possible fractionation processes occurring on the way from the soil to the cave, comparative studies of LOPs in soil, drip water and speleothems should be carried out and complemented with other methods, such as total organic carbon mesasurements, the calculation of C/N ratios or fluorescence spectroscopy. Annually laminated speleothems might be well suited as study objects because they often possess high organic matter content and high seasonality. In addition to stalagmites, flowstones could be valuable sample material because they are often fed by water flows with higher discharge carrying larger amounts of organic material.

17. P19, L2: to be mitigate
See reply to comment 16.

18. P19, L5–6: but may be linked to microorganism in the soils...
See reply to comment 16.

19. P19, L12–13: and complete with total organic carbon measurement... together with C/N ration for instance...
See reply to comment 16.

[revised manuscript text omitted]

---

## Author Response (AR1)

Dear Dr Erin McClymont,

thank you for carefully evaluating our manuscript. We have tried to address your points by adding or changing text in several parts of the manuscript. The line and page references refer to the new manuscript uploaded today. In addition, we have attached a marked-up version of the manuscript to this document highlighting the changes made to the text.

(1) You have requested clarification of where the alteration of the original lignin signal might occur in the environment
- In the abstract, we have slightly changed the line you cited to avoid misinterpretation. P. 1, l. 3: "Lignin is only produced by vascular plants and therefore has the potential to be an unambiguous vegetation proxy and to complement other vegetation and climate proxies in speleothems."
- In the discussion part, we have included a more detailed discussion of where and how the C/V and S/V ratios can be altered P. 15, l. 21-33:
  "However, biotic factors in the soil, such as different rates of microbial degradation for the different types of lignin, as well as transport factors in the soil or the aquifer could also have an influence on the C/V and S/V ratios. Degradation studies of leaf litter showed that the microbial degradation of lignin high in C- and S-group monomeric units is faster than for lignin high in V-group units due to structural differences between the monomers (Bahri et al., 2006; Opsahl and Benner, 1995; Jex et al., 2014). In addition, adsorption of lignin to mineral particles can lead to an increase in the C/V and S/V ratios, indicating a preferential adsorption of lignin high in V-group monomeric units (Hernes et al., 2007, 2013). This may happen both in the soil and the aquifer. We assume that more degraded lignin is present in smaller fragments and a higher oxidation state and should therefore be better soluble in water. In addition, we hypothesize that this could lead to a more efficient transport through the soil and the aquifer to the cave. This could explain the increase of C/V and S/V ratios during drier times. In addition, the transport of lignin is also likely to involve processes such as coiling with humic substances in the soil. Currently, we cannot quantify the contribution of the individual mentioned processes to the final lignin signals in the drip water and speleothem calcite. To further investigate all these influencing factors, comparative LOP studies of soil, drip water and speleothem samples need to be conducted."

(2) You have requested a clarification of how the drip water relates to the speleothem data
- Your interpretation of our statement was exactly what we wanted to convey. We have tried to make our statement clearer now. P. 17, l. 7-15:
  "The difference in seasonal variations of Σ8 and phosphate could be explained either by different sources, as phosphate can also be leached from the bedrock, or by different transport mechanisms, such as transport in colloids or in solution. It is important to note that we are looking at the seasonal timescale here. On seasonal timescales, the variation of the phosphate concentration in the drip water is probably mainly influenced by direct contribution from the bedrock by different water flows, while the input of plant material from the reservoir of organic matter can be considered as relatively constant. On longer timescales with a resolution of decades to centuries, as recorded in the stalagmite, the link between phosphorous concentrations and the general activity of the vegetation may be stronger. This is because the release and mobilization of phosphorous from the bedrock through weathering is related to the productivity of the vegetation above the cave since

phosphorous serves as a plant nutrient."

(3)  P. 2, l. 10-11: The question mark in "Lipid biomarkers, such as fatty acids (?) and…" has been replaced by the missing literature reference (Bosle et al., 2014).

(4)  P. 12, l. 10-12 and l. 30: All "??" have been replaced by the respective table numbers of the supplementary information.

(5)  The title has been changed to "Lignin oxidation products as a potential proxy for vegetation and environmental changes in speleothems and cave drip water – a first record from the Herbstlabyrinth, central Germany"

(6)  In addition, we have moved all tables from the appendix of the manuscript to the supplementary information.

[revised manuscript text omitted]

Pearson's correlation coefficients $r$ (with $p < 0.05$) of the middle part of the stalagmite Pearson's $r$ C/V - 0.82 -0.51 - - - -0.48 - -0.56 - S/V 0.82 - -0.54 - - - -0.53 - -0.57 - -0.45 Σ8 -0.51 -0.54 - - - - 0.66 - 0.60 - 0.79 $\delta^{13}$C - - - - 0.53 0.69 -0.65 - - -0.59 - $\delta^{18}$O - - - 0.53 - 0.50 - - - -0.64 - Mg - - - 0.69 0.50 - - - - -0.55 - P -0.48 -0.53 0.66 -0.65 - - - - 0.73 0.80 0.56 Sr - - - - - - - 0.67 - - Ba -0.56 -0.57 0.60 - - - 0.73 0.67 - 0.78 - U - - - - -0.59 -0.64 -0.55 0.80 - 0.78 - - growth rate - -0.45 0.79 - - - - 0.56 - - - - -

Spearman's correlation coefficients $\rho$ (with $p < 0.05$) of the middle part of the stalagmite Spearman's $\rho$ C/V - 0.80 -0.53 - - - - - -0.57 - - S/V 0.80 - - - - - - -0.45 - - -0.53 - - Σ8 -0.53 - - - - - 0.52 - 0.63 - - $\delta^{13}$C - - - - 0.61 0.56 -0.64 - - - -0.63 - $\delta^{18}$O - - - 0.61 - 0.60 - - - - -0.55 - Mg - - - - 0.56 0.60 - - - - -0.54 - P - -0.45 0.52 -0.64 - - - - 0.60 0.75 - Sr - - - - - - - - - 0.73 - - Ba -0.57 -0.53 0.63 - - - 0.60 0.73 - 0.76 - U - - - - -0.63 -0.55 -0.54 0.75 - 0.76 - - growth rate - - - - - - - - - - -

Pearson's correlation coefficients $r$ (with $p < 0.05$) of the older part of the stalagmite Pearson's $r$ C/V - 0.70 - - - - -0.52 - -0.77 0.62 0.61 - S/V 0.70 - - 0.52 0.56 -0.54 - -0.71 0.72 0.58 - Σ8 - - - - - - - - - - - - $\delta^{13}$C - 0.52 - - 0.67 -0.57 - - - 0.53 - - 0.60 $\delta^{18}$O - 0.56 - 0.67 - -0.56 - - - 0.65 - -0.65 Mg -0.52 -0.54 - -0.57 -0.56 - -0.59 0.85 -0.92 -0.79 - P - - - - - - -0.59 - -0.59 - 0.83 - Sr -0.77 -0.71 - - - - 0.85 -0.59 - -0.89 -0.89 - Ba 0.62 0.72 - 0.53 0.65 -0.92 - -0.89 - 0.81 - U 0.61 0.58 - - - - -0.79 0.83 -0.89 0.81 - - growth rate - - - - -0.60 -0.65 - - - - - - -

Spearman's correlation coefficients $\rho$ (with $p < 0.05$) of the older part of the stalagmite Spearman's $\rho$ C/V - 0.76 - - - - -0.71 - -0.77 0.66 0.71 - S/V 0.76 - - 0.45 - -0.68 - -0.77 0.71 0.67 - Σ8 - - - - - - - - - - - - $\delta^{13}$C - 0.45 - - 0.58 - - - - 0.49 - -0.75 $\delta^{18}$O - - - 0.58 - -0.49 - - 0.65 - -0.78 Mg -0.71 -0.68 - - - -0.49 - -0.45 0.93 -0.88 -0.88 P - - - - - - -0.45 - -0.48 - 0.72 - Sr -0.77 -0.77 - - - -0.93 -0.48 - -0.85 -0.88 - Ba 0.66 0.71 - 0.49 0.65 -0.88 - -0.85 - 0.85 - -0.62 U 0.71 0.67 - - - - -0.88 0.72 -0.88 0.85 - - growth rate - - - - -0.75 -0.78 - - - - 0.62 - -

Coefficients of the principal component analysis of the stalagmite Coefficients of C/V 0.31 -0.42 -0.10 0.08 -0.31 -0.26 -0.21 0.46 0.30 0.23 -0.38 0.34 S/V 0.29 -0.44 -0.01 0.13 -0.33 -0.23 -0.08 0.40 0.33 -0.06 -0.19 0.54 S8 -0.01 0.44 -0.14 0.44 0.32 0.30 -0.11 0.22 -0.04 -0.09 0.72 0.61 d13C 0.15 -0.20 0.45 0.44 -0.30 0.26 0.39 0.35 0.25 -0.42 0.02 0.08 d18O -0.08 0.14 0.59 0.22 -0.14 0.53 0.12 -0.25 0.26 -0.43 -0.07 -0.07 Mg -0.40 -0.13 -0.35 0.15 -0.23 0.40 -0.03 0.37 -0.39 0.02 -0.18 0.21 P 0.34 0.35 -0.18 -0.15 0.40 -0.01 -0.27 0.17 0.23 0.38 0.45 -0.27 Sr -0.48 0.15 0.09 -0.13 0.20 -0.07 0.70 0.14 -0.39 -0.19 0.08

0.03 Ba 0.25 0.36 0.36 -0.13 0.39 0.03 0.29 0.33 0.41 -0.05 -0.01 -0.09 U 0.45 0.18 -0.13 -0.26 0.36 -0.28 0.00 0.22 0.36 0.27 0.18 -0.10 growth rate 0.15 0.23 -0.33 0.62 0.22 0.44 -0.35 0.23 -0.12 0.56 -0.15 0.26

Eigenvalues and explained variance of the principal component analysis of the stalagmite 1 3.47 31.55% 31.55% 4.95 44.99% 44.99% 5.63 51.14% 51.14% 2 2.96 26.88% 58.42% 2.45 22.27% 67.26% 2.24 20.36% 71.50% 3 2.07 18.81% 77.23% 1.39 12.64% 79.90% 1.26 11.44% 82.95% 4 1.12 10.21% 87.44% 0.95 8.67% 88.57% 0.82 7.41% 90.36% 5 0.36 3.29% 90.73% 0.54 4.92% 93.49% 0.36 3.27% 93.63% 6 0.31 2.83% 93.56% 0.28 2.54% 96.03% 0.28 2.55% 96.19% 7 0.25 2.26% 95.82% 0.17 1.51% 97.54% 0.20 1.80% 97.98% 8 0.21 1.92% 97.73% 0.14 1.30% 98.85% 0.15 1.37% 99.36% 9 0.13 1.16% 98.90% 0.06 0.53% 99.38% 0.04 0.34% 99.70% 10 0.08 0.77% 99.66% 0.05 0.44% 99.81% 0.02 0.22% 99.92% 11 0.04 0.34% 100.00% 0.02 0.19% 100.00% 0.01 0.08% 100.00%

[revised manuscript text omitted]